# POLRMT mutations impair mitochondrial transcription causing neurological disease

Monika Oláhová[1,26], Bradley Peter[2,26], Zsolt Szilagyi[2], Hector Diaz-Maldonado[2], Meenakshi Singh[2], Ewen W. Sommerville[1], Emma L. Blakely[1], Jack J. Collier[1], Emily Hoberg[2], Viktor Stránecký[3], Hana Hartmannová[3], Anthony J. Bleyer[3,4], Kim L. McBride [5], Sasigarn A. Bowden [6], Zuzana Korandová[3,7], Alena Pecinová [7], Hans-Hilger Ropers[8,9], Kimia Kahrizi[10], Hossein Najmabadi[10], Mark A. Tarnopolsky[11], Lauren I. Brady[11], K. Nicole Weaver[12,13], Carlos E. Prada[12,13,14], Katrin Õunap [15,16,17], Monica H. Wojcik[17,18], Sander Pajusalu [15,16,19], Safoora B. Syeda[20], Lynn Pais[21], Elicia A. Estrella[22], Christine C. Bruels[20], Louis M. Kunkel[22], Peter B. Kang [20,23,24], Penelope E. Bonnen[25], Tomáš Mráček[7], Stanislav Kmoch[3], Gráinne S. Gorman[1], Maria Falkenberg [2], Claes M. Gustafsson [2,27✉] & Robert W. Taylor [1,27✉]

While >300 disease-causing variants have been identified in the mitochondrial DNA (mtDNA) polymerase γ, no mitochondrial phenotypes have been associated with POLRMT, the RNA polymerase responsible for transcription of the mitochondrial genome. Here, we characterise the clinical and molecular nature of POLRMT variants in eight individuals from seven unrelated families. Patients present with global developmental delay, hypotonia, short stature, and speech/intellectual disability in childhood; one subject displayed an indolent progressive external ophthalmoplegia phenotype. Massive parallel sequencing of all subjects identifies recessive and dominant variants in the POLRMT gene. Patient fibroblasts have a defect in mitochondrial mRNA synthesis, but no mtDNA deletions or copy number abnormalities. The in vitro characterisation of the recombinant POLRMT mutants reveals variable, but deleterious effects on mitochondrial transcription. Together, our in vivo and in vitro functional studies of POLRMT variants establish defective mitochondrial transcription as an important disease mechanism.

A list of author affiliations appears at the end of the paper.

Mitochondrial diseases comprise a wide spectrum of genetic disorders with varied clinical features and patient prognoses[1]. These often multi-organ disorders can be sporadic or due to inherited pathogenic variants in either mitochondrial DNA (mtDNA) or in the ~1200 nuclear-encoded genes required for mitochondrial function and maintenance. These mtDNA maintenance proteins include the DNA polymerase γ (POLγ) and TWINKLE helicase required for replication of mtDNA. In addition, priming of mtDNA replication as well as oxidative phosphorylation (OXPHOS) system gene expression is known to be reliant upon the mtDNA transcription machinery[2–4]. Transcription of mtDNA is carried out by the mitochondrial RNA polymerase (POLRMT) and the mitochondrial transcription factors A (TFAM) and B2 (TFB2M) that initiate promoter-specific transcription from the light-strand and heavy-strand promoters (LSP and HSP)[5–7]. During replication of mtDNA, POLRMT also synthesises the RNA primers required for initiation from the two mitochondrial origins of replication, OriL and OriH[4,8].

Structurally, POLRMT comprises four main domains: the N-terminal extension, a pentatricopeptide repeat (PPR) domain, the N-terminal domain, and the C-terminal domain[9]. Large-scale conformational changes occur in these domains as the protein binds TFAM and mtDNA during transcription initiation as well as during the handover to processive mtDNA transcription[9–12]. The initiation of transcription begins with TFAM binding the promoter upstream of the transcription start site and inducing a stable U-bend in the mtDNA. Following the recruitment of POLRMT a closed pre-initiation complex is formed. The interaction between POLRMT with TFAM is mediated via the N-terminal extension domain of POLRMT, also known as the 'tether helix', which is important for anchoring the active site of POLRMT near the transcription start site where the initial DNA melting occurs[11]. Following the binding of TFB2M to the duplex DNA, TFB2M induces structural changes in POLRMT that stabilise the open DNA region. TFB2M contacts the conserved intercalating hairpin within the N-terminal domain of POLRMT, that results in melting of the DNA duplex and formation of the open initiation complex, where de novo RNA synthesis can begin[11]. The transition from transcription initiation to elongation involves the release of TFB2M and recruitment of the mitochondrial transcription elongation factor (TEFM). TEFM forms a homodimer via its C-terminus and stabilises interactions between the intercalating hairpin of POLRMT and the DNA template, thus separating the nascent RNA[13–15]. This interaction also contributes to the formation of a tight RNA exit tunnel along POLRMT, that is thought to enhance the processivity of the RNA polymerase along the DNA strand, and bypass premature termination or stalling caused by secondary structured RNA regions such as the G-quadruplex sequence of the conserved sequence block II (CSBII)[13–15]. In the absence of TEFM, the LSP-driven transcription is often terminated at the CSBII producing a 120-nt transcript, which after processing can be used by the replisome machinery to initiate replication at OriH[4].

The vast majority of mitochondrial disorders result from defects in components of the mtDNA maintenance machinery and the OXPHOS system[16,17]. Variants in POLG and TWNK, the genes encoding POLγ and TWINKLE, represent a frequent cause of inherited mitochondrial disease, with more than 300 different disease-causing variants having been identified in POLG alone (https://tools.niehs.nih.gov/polg/). These mutations lead to a disorder of mtDNA maintenance which can manifest as mtDNA depletion or variable mtDNA deletions[18]. In contrast, pathogenic variants in POLRMT have not, until now, been associated with mitochondrial disease.

Here we report a cohort of eight patients with POLRMT mutations associated with mitochondrial dysfunction and a broad spectrum of neurological presentations which are independent of disorders of mtDNA maintenance. Respiratory chain deficiencies in cultured patient fibroblasts are linked to the quantitative loss of OXPHOS mRNA transcripts in vivo whereas mtDNA depletion/deletions were absent. In vitro assessment of the POLRMT variants reveals a negative effect on processive transcription activity and a reduced ability to synthesise primers for mtDNA replication. Our data demonstrate that defective POLRMT can lead to classical mitochondrial disease and wider neurological manifestations which are independent of disordered mtDNA maintenance.

## Results

**Identification of patients with pathogenic POLRMT variants.** We identified 8 individuals from 7 families (Fig. 1a, Supplementary results and Tables 1, 2) with a broad range of neurological manifestations and rare POLRMT variants using a range of massive parallel sequencing strategies. The most frequent clinical presentation of our patients was developmental delay including fine motor, gross motor, language, social/behavioural, and thinking/intellectual (five patients). This was often accompanied by low BMI (five patients), hypotonia (four patients), gastrointestinal (three patients), limb-girdle weakness (three patients), short stature (four patients), and eye movement abnormalities (four patients); resulting in mild to severe disability. Other muscle symptoms such as facial weakness (two patients), contractures (one patient), and neck ptosis (one patient) were less frequent. Other symptoms were relatively uncommon and included electrolyte abnormalities (three patients), skeletal abnormalities (two patients), dysmorphism (two patients), renal dysfunction (two patients), anaemia (two patients), sensorineural hearing loss (one patient), and epilepsy (one patient). Cardiac involvement was conspicuously absent. All POLRMT variants were confirmed using Sanger sequencing and phase established confirming the inheritance pattern where possible (Fig. 1a). Detailed clinical and molecular genetic findings are provided in Tables 1, 2, and the supplementary appendix.

In Patient 1 (P1) (Fig. 1a and Table 1), whole exome sequencing (WES) of all four family members led to the identification of three POLRMT variants in the proband (NM_005035.3, https://www.ncbi.nlm.nih.gov/nuccore/NM_005035.3); two paternally-inherited, in cis, missense variants c.1696C > T (p.Pro566Ser) and c.3578C > T (p.Ser1193Phe) and a maternally-inherited c.2608G > A (p.Asp870Asn) variant. In gnomAD (https://gnomad.broadinstitute.org/), the p.Pro566Ser variant was found in 401/200344 alleles (minor allele frequency (MAF), $2.0 \times 10^{-3}$) and the p.Ser1193Phe variant was found in 452/279886 ($1.61 \times 10^{-3}$) alleles, all in heterozygous state. Both missense variants were predicted to be benign by PolyPhen-2 (http://genetics.bwh.harvard.edu/pph2/). The maternal variant p.Asp870Asn (rs139383492) was found in 992/275368 alleles ($3.6 \times 10^{-3}$) in gnomAD, which included 7 homozygotes. The variant was predicted as 'possibily damaging' in PolyPhen-2.

In Patient 2 (P2), analysis using a 406 custom, nuclear-encoded mitochondrial gene panel identified two rare POLRMT variants, c.2225_2242del (p.Pro742_Pro747del) and c.748C > G (p.His250Asp) (Fig. 1a and Table 1). The p.Pro742_Pro747del in-frame deletion was found in 8/47292 alleles ($1.69 \times 10^{-4}$) and the p.His250Asp variant was present in 50/276422 alleles ($1.81 \times 10^{-4}$) in gnomAD, all in heterozygous state. Sanger sequencing confirmed that both variants were in trans, establishing recessive inheritance (Fig. 1a).

In Patient 3 (P3), WES identified a rare heterozygous c.2641-1G > C POLRMT splice-acceptor variant in intron 10, which was confirmed by Sanger sequencing (Fig. 1a and Table 1). Parental and familial segregation studies were not possible. The variant

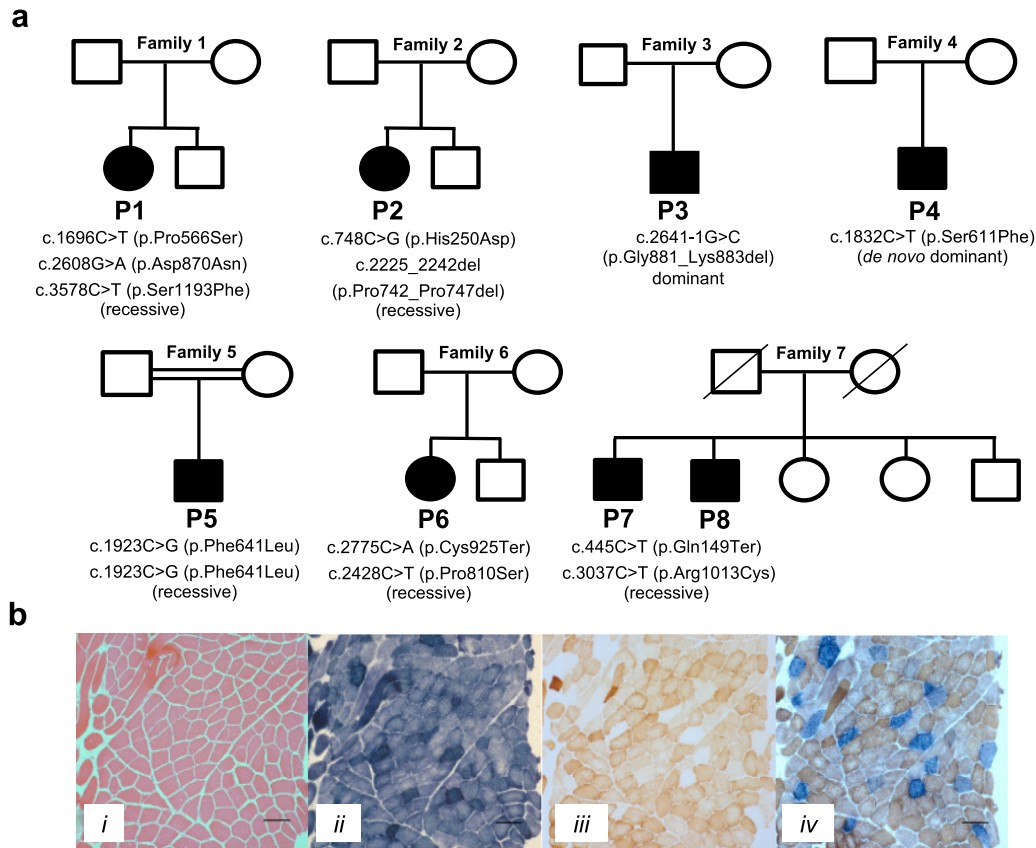

**Fig. 1 Molecular genetics and histochemical studies. a** Family pedigrees and segregation status of *POLRMT* patients. Affected individuals are depicted in black, circles represent females, squares represent males; a diagonal line through the symbol represents deceased individuals and double horizontal lines indicate consanguinity. **b** Diagnostic histochemical analyses ($n = 2$) of skeletal muscle-derived from P3 showing (i) Haematoxylin & Eosin staining (ii) SDH (iii) COX and (iv) COX-SDH reactivities. The sequential COX-SDH reaction in (iv) revealed a mosaic pattern of COX-deficiency in P3. The scale bar shown is 100 um.

was absent from in-house controls ($n = 378$) but was present in heterozygous state in gnomAD in two Non-Finnish Europeans ($9.49 \times 10^{-6}$). No pathogenic or likely pathogenic variants were identified in known nuclear genes associated with mtDNA maintenance disorders. No additional rare variants in *POLRMT* were identified and analysis of exome sequencing copy number variants did not disclose rearrangements encompassing *POLRMT*. Further to WES, whole genome sequencing of Patient 3 failed to identify a second *POLRMT* variant or any other likely genetic cause. To assess the potential effect on exon splicing, total RNA was extracted from emetine-treated and untreated fibroblasts from P3 and a control. Following RT-PCR, sequencing of fibroblast-derived cDNA products showed that the c.2641-1G > C variant disrupts the consensus splice-acceptor site of intron 10, leading activation of a cryptic splice-site in exon 11 of *POLRMT* and removal of nucleotides r.2641_2649 from the transcript (Supplementary Figure S1). The predicted effect at protein level is an in-frame deletion of three amino acids p.Gly881_Lys883del. The presence of this mutant transcript in both emetine-treated and untreated patient fibroblasts confirmed that it is not subjected to nonsense-mediated decay.

Patient 4 (P4) was subject to diagnostic trio WES after candidate gene screening for possible causal variants in genes associated with hypomagnesemia (*CLDN16*, *CLDN19*, *CNNM2*, *EGF*, *FXYD2*, *KCNA1*, *SLC12A3* and *TRPM6*), which identified only a single possible pathogenic *SLC12A3* variant (XomeDx Slice, Genedx). Given the presence of hypercalciuria was not consistent with Gittelman syndrome, deletion/duplication testing of *SLC12A3* was not pursued. The trio WES strategy identified a

de novo c.1832C > T (p.Ser611Phe) *POLRMT* gene variant (Fig. 1a and Table 1). This variant was absent from both gnomAD and the 1000 Genomes Consortium (http://www.internationalgenome.org/home). MutationTaster, Provean, and SIFT all predict the variant to be damaging.

Trio WES strategy was also employed in the family of Patient 5 (P5), leading to identification of a rare homozygous missense *POLRMT* variant c.1923C > G (p.Phe641Leu). The Phe641 amino acid was completely conserved from human to *Drosophila* and *C. elegans*, with high in silico pathogenicity prediction scores (Fig. 1a and Table 1). The parents of P5 were confirmed to be heterozygous carriers. The p.Phe641Leu variant was found in 1/237432 alleles ($4.21 \times 10^{-6}$) in gnomAD and was absent from the Iranome Consortium (http://www.iranome.ir/).

Trio WES analysis in the family of Patient 6 (P6) identified two *POLRMT* variant alleles, a maternally-inherited c.2775C > A (p.Cys925*) nonsense variant and a paternally-inherited c.2428C > T (p.Pro810Ser) missense variant (Fig. 1a and Table 1). The p.Cys925* variant was found in 4/31368 alleles ($1.28 \times 10^{-4}$) in GnomAD, all in heterozygous state. By contrast, the p.Pro810Ser variant was found in 386/264146 alleles ($1.46 \times 10^{-3}$) in gnomAD, which included four homozygotes. Nevertheless, the p.Pro810Ser variant is predicted to be (possibly) damaging by various in-silico algorithms like PolyPhen-2 (0.45), SIFT (0.04), MutationTaster, and CADD (score 24.4).

Genetic testing for limb-girdle muscular dystrophy (LGMD) genes and a neuromuscular disease genetic panel were initially performed on Patient 7 (P7) and Patient 8 (P8) respectively. However, no pathogenic variants were found. Next, WES was

**Table 1 Genetic findings in individuals with POLRMT variants.**

| ID | cDNA (NM_005035.3) protein (NP_005026.3) | Minor Allele Frequency (gnomAD) | CADD score | Age-at-onset/Clinical course | Ethnicity |
|---|---|---|---|---|---|
| Family 1 Patient 1[a] female | c.1696C > T, p.Pro566Ser | $2.0 \times 10^{-3}$ | 14.93 | 1y/Alive 16y | Dominican Mixed Ancestry |
| | c.2608G > A, p.Asp870Asn | $3.6 \times 10^{-3}$ | 16.0 | | |
| | c.3578C > T, p.Ser1193Phe (recessive) | $1.61 \times 10^{-3}$ | 15.17 | | |
| Family 2 Patient 2[a,b] female | c.748C > G, p.His250Asp | $1.81 \times 10^{-4}$ | 21.7 | Birth/Alive 19y | Northern European |
| | c.2225_2242del, p.Pro742_Pro747del (recessive) | $1.69 \times 10^{-4}$ | n/a | | |
| Family 3 Patient 3[a,c] male | c.2641-1G > C, p.Gly881_Lys883del (dominant) | $9.49 \times 10^{-6}$ | n/a | 28y/Alive 57y | British |
| Family 4 Patient 4[a] male | c.1832C > T, p.Ser611Phe (de novo dominant) | Ø | 25.3 | 10m/Alive 3y | North American |
| Family 5 Patient 5[a] male | c.1923C > G, p.Phe641Leu (recessive) | $4.21 \times 10^{-6}$ | 16.02 | 1y/Alive 10y | Iranian[d] |
| Family 6 Patient 6[a] female | c.2775C > A, p.Cys925* | $1.28 \times 10^{-4}$ | 53 | Birth/Alive 14y | Northern European |
| | c.2428C > T, p.Pro810Ser (recessive) | $1.46 \times 10^{-3}$ | 24.8 | | |
| Family 7 Patient 7[a] male | c.445C > T, p.Gln149* | $4.01 \times 10^{-6}$ | 35 | Early | Italian ancestry |
| | c.3037C > T, p.Arg1013Cys (recessive) | Ø | 29.4 | 40's/Alive 59y | |
| Family 7 Patient 8[a] male | c.445C > T, p.Gln149* | $4.01 \times 10^{-6}$ | 35 | Early | Italian ancestry |
| | c.3037C > T, p.Arg1013Cys (recessive) | Ø | 29.4 | 40's/Alive 72y | |

[a]investigated by whole-exome sequencing;
[b]investigated by mitochondrial gene panel;
[c]investigated by whole genome sequencing;
[d]consanguineous parents; Ø – variant absent in gnomAD.

performed in both P7 and P8, leading to the identification of two heterozygous *POLRMT* variants present in both probands, a c.445C > T (p.Gln149*) nonsense variant and c.3037C > T (p.Arg1013Cys) missense variant (Fig. 1a, Table 1). Parental segregation analysis was not possible. Nonetheless, segregation studies confirmed both variants in P7 and P8, while an unaffected sibling was heterozygous for only the p.Arg1013Cys variant. In gnomAD, the p.Gln149* variant was present in 1/249286 alleles ($4.1 \times 10^{-6}$). By contrast, the p.Arg1013Cys variant is absent from gnomAD.

**Histological and biochemical studies.** Muscle biopsies were available from four patients (P1, P2, P3, P7) and these were subjected to routine, diagnostic histopathological investigations (Table 2). In P1, increased levels of neutral lipid were observed in the myofibers. Histologically, two muscle biopsies performed in P2 at 8 and 12 years of age only showed non-specific changes. P3 showed a mosaic pattern of cytochrome *c* oxidase (COX)-deficiency following sequential COX-SDH (succinate dehydrogenase) histochemistry (Fig. 1b), in addition to occasional ragged-red fibres. A muscle biopsy available from P7 showed variation of fibre size with type 2 fibre predominance and rare "nuclear bags". Electron microscopy (EM) showed subsarcolemmal accumulation of mitochondria with moderate pleomorphism, along with rectangular crystalline inclusions present in mitochondria.

Spectrophotometric assessment of mitochondrial OXPHOS complex activities in the muscle from P1 showed a respiratory chain defect in Complex I associated with a mild loss of Complex III activity[19]. Biochemical assessment of the mitochondrial respiratory chain complex activities and the matrix marker citrate synthase in P2 skeletal muscle homogenate revealed a defect in Complexes I + III, while Complex IV and Complexes II + III activities were reported to be similar to controls (Table 2).

**Structural analysis of POLRMT variants.** Assessment of the structural implications of the patient variants carried out in silico predicted that all of the identified mutations could have negative effects on transcription initiation or elongation. Although several of the identified variants cluster in the catalytic C-terminal domain, mutations also occur in the PPR and N-terminal domains (Fig. 2a). All residues appear to be involved in either (1) intramolecular interactions which stabilise the core structure of POLRMT (P1, P2, P5, P6, P7, P8); (2) conformational flexibility required for the switch from transcription initiation to elongation (P3); (3) protein-protein interactions which facilitate cofactor binding (P4); or (4) protein–DNA interactions which facilitate transcription (P2, P4) (Fig. 2b, c). P1 harbours the compound heterozygous variants p.Asp870Asn (D870N) and p.Pro566Ser (P566S)/p.Ser1193Phe (S1193F). P566 is located between two helices (α14–α15) and induces a kink in the succeeding helix α15, whereas D870 forms a stabilising salt-bridge with R882 located on an adjacent loop (Fig. 2b). S1193 protrudes into the solvent and is surrounded by predominantly bulky residues (K1189, R1190, and F1191). The His250Asp (H250D) POLRMT variant present in P2 is located in close proximity to the TFAM binding site and forms a stabilising interaction with Y264 and D289 (Fig. 2b). P2 also harbours a deletion of residues 742–747, a region located in close proximity to the bound promoter DNA (Fig. 2b). P3 harbours a deletion of residues 881–883 (Δ881–3) in the palm subdomain. The loop containing residues 881–883 is stably associated with an adjacent helix by several H-bonds (Fig. 2b). This region is conserved in the T7 RNA polymerase and subsequent mutations would likely perturb the integrity of this structural element. P4 harbours the Ser611Phe (S611F) variant which is located on the intercalating hairpin in

**Table 2 Clinical and pathological findings in individuals with *POLRMT* variants.**

| ID | Clinical features and relevant clinical investigations | Muscle biopsy findings | Biochemical and laboratory findings |
|---|---|---|---|
| Family 1 Patient 1[a] female | GA: 36 weeks, birth weight 4100 grams, birth length 50 cm, OFC 38 cm; Genu varum around age 1 year; proximal tubulopathy or renal Fanconi syndrome with hypophosphatemic rickets; mild developmental delays and hypotonia; short stature; BMI 27 kg/m$^2$; elevated lactate; enuresis; normal pubertal development; stage 3 chronic kidney disease; white matter volume loss with mild ventriculomegaly and thinning of the corpus callosum | Increased lipid in the myofibres | Mild combined CI + CIII respiratory chain defect in fibroblasts. CI deficiency and milder loss of CIII activity in skeletal muscle mtDNA testing for point mutations, deletions and CI gene sequencing normal |
| Family 2 Patient 2[a,b] female | Intellectual disability; high frequency hearing loss; BMI 28.4 kg/m$^2$; developmental delay; mild hypotonia; bilateral ptosis; left eye exotropia; constipation; normal lactate and pyruvate levels; several foci of FLAIR high signal in subcortical white matter of the frontal lobes bilaterally; mild increase in lactate on MRS in the left insular cortex and thalamus | Mitochondrial aggregates on EM, normal biopsy histologically | Mild combined CI + CIII respiratory chain defects in fibroblasts and skeletal muscle No evidence of mtDNA depletion, single or multiple mtDNA deletions in muscle |
| Family 3 Patient 3[a,c] male | Indolent PEO phenotype; bilateral ptosis; brain MRI is normal; an uncle was reported to have PEO but this was never clinically-confirmed | 25% COX-deficient muscle fibres, some ragged-red fibres | Mild CI and CIV defects in skeletal muscle No evidence of mtDNA copy number abnormality, mtDNA rearrangements or mtDNA point mutations |
| Family 4 Patient 4[a] male | Birth weight 3315 grams, length 52 cm, OFC 36.5 cm; at the age 3.5 y: OFC 51 cm (75th centile), height 83 cm (<3rd centile), weight 12.3 kg (10th centile); global, mild developmental delay; hypotonia; expressive-receptive language difficulties (20 words at 3.5 years); short stature; low phosphorous; medullary nephrocalcinosis; one episode of anaemia requiring a blood | Not determined | Respiratory chain studies not undertaken Renal hypomagnesemia: 0.6 mg/dl (normal range 1.7–2.4 mg/dl) Elevated lactate: 2.4 to 3.8 mmol/l (normal range 0.7–2.1 mmol/l) Normal creatine kinase levels |
| Family 5 Patient 5[a] male | GA: 39 weeks, weight 2800 grams, length 48 cm, OFC 31 cm; profound intellectual disability (IQ 20); microcephaly; severe global developmental delay (head control:13 m, sitting: 4 y, walking 8 y); short stature; strabismus; no speech; generalised myopathy; walking with assistance | Not determined | Respiratory chain studies not undertaken |
| Family 6 Patient 6[a] female | GA: 37 weeks, weight 2690 grams (−1 SD), length 47 cm (−1 SD), OFC 32 cm (−1.5 SD); bilateral club foot; epicanthal folds; strabismus; hypotelorism; blue scleras; low nasal bridge; upturned nose; opened mouth appearance; high palate; dysmorphic ears; bilateral simian crease; no sucking reflex in infancy; moderate intellectual disability; microcephaly; global developmental delay; short stature; strabismus; no speech; focal epilepsy; muscular hypotonia; iron deficient anaemia and thrombocytopenia; secondary carnitine deficiency; low folate; MRI brain: ventriculus septi pellucidi | Not determined | Unspecific organic aciduria (increased excretion of Krebs cycle metabolites and methylmalonate) Lactate/pyruvate ratio: 70 (normal range <25) |
| Family 7 Patient 7[a] male | Predominant muscle weakness, accompanied by muscle atrophy | Myopathic findings on electromyography - type 2 fibre predominance, rare "nuclear bags", some COX-deficient fibres, mitochondrial abnormalities on EM | Respiratory chain enzymes and coenzyme Q10 levels normal in skeletal muscle Elevated creatine kinase levels: 873 U/L (normal range 175 U/L) |
| Family 7 Patient 8[a] male | Proximal muscle weakness, neurological examination is notable for right exotropia | Myopathic findings on electromyography | Mildly elevated creatine kinase levels: 307 U/L (normal 175 U/L) |

[a]investigated by whole-exome sequencing;
[b]investigated by mitochondrial gene panel;
[c]investigated by whole-genome sequencing; GA gestational age; OFC occipital frontal circumference; BMI body mass index; MRI magnetic resonance imaging; MRS magnetic resonance spectroscopy; CI Complex I; CIII Complex III; EM electron microscopy, n/a not applicable.

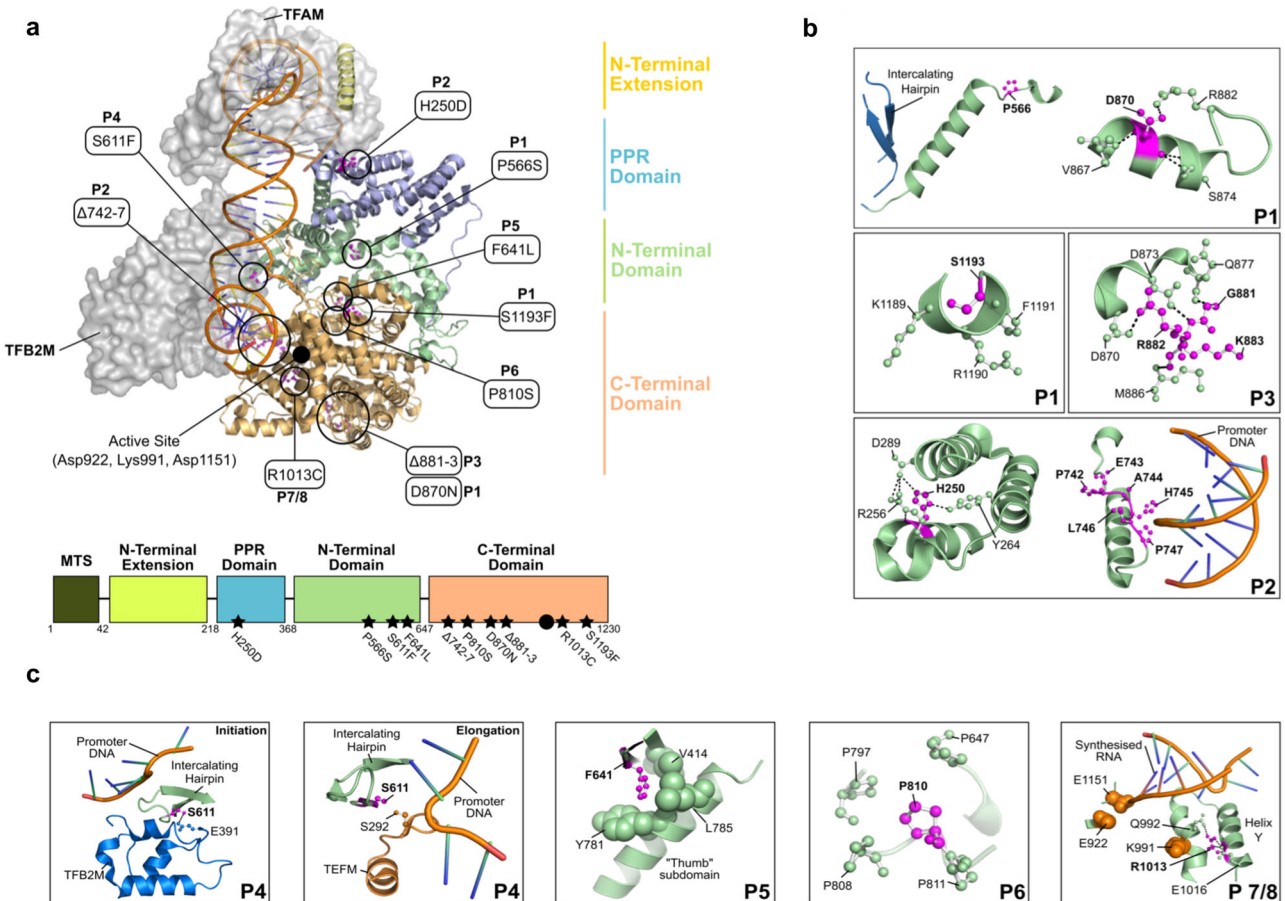

**Fig. 2 Domain organisation of POLRMT. a** The structure of the transcription initiation complex is shown bound to the heavy strand promoter (HSP) (PDB ID: 6ERQ). POLRMT comprises four main domains: the N-terminal extension (yellow), a pentatricopeptide repeat (PPR) domain (purple), the N-terminal domain (green), and the C-terminal domain (orange). The positions of the disease-causing variants analysed in this study are indicated (black circles) as well as the location of the active site (full black circle). **b, c** Interactions formed between the residues implicated in disease (magenta, bold text) and surrounding residues (green). **b** P1 harbours the compound heterozygous variants D870N and P566S/S1193F, P2 harbours the H250D variant as well as a deletion of residues 742–747 and P3 harbours a deletion of residues 881–883. **c** P4 harbours the S611F variant, P5 harbours the F641L variant, P6 harbours the P810S variant, and P7/8 harbour the R1013C variant. Structures in (**b, c**) are generated from the transcription initiation complex (PDB ID: 6ERP) and transcription elongation complex (PDB ID: 5OLA).

close proximity to the promoter DNA (Fig. 2c). Interestingly, this residue forms a hydrogen bond with E391 of the TFB2M cofactor in the transcription initiation complex (Fig. 2c) and is also located in close proximity to both the elongation factor TEFM and promoter DNA in the transcription elongation complex (Fig. 2c). In P5, the homozygous p.Phe641Leu (F641L) variant fits tightly into a hydrophobic pocket (Fig. 2c), whereas P6 harbours a p.Pro810Ser (P810S) variant which is located in close proximity (~7 Å) to the DNA template. Interestingly, P810 is located at the centre of a stabilising proline cluster, the residues of which are highly conserved across both vertebrate and invertebrate species (Fig. 2c). Finally, P7 and P8 harbour an p.Arg1013Cys (R1013C) variant which is located on helix Y within the active site and in close proximity to the catalytic residues (Fig. 2c). The movement of this helix is associated with nucleotide addition and translocation along the DNA. In the transcription elongation complex, R1013 forms a salt bridge with E1016 and a hydrogen bond with Q992 (Fig. 2c).

**Patient _POLRMT_ variants do not cause mtDNA depletion/ deletions but are associated with lower mRNA transcript levels in vivo.** The mtDNA molecular analysis for point mutations and deletions was normal in P1 muscle as previously reported[19].

Diagnostic analysis of P2 total muscle DNA homogenate showed no evidence of mtDNA copy number abnormalities, single or multiple mtDNA deletions using long-range PCR, while full mtDNA sequencing failed to detect rare mtDNA variants other than mtDNA haplogroup markers. Similarly, mitochondrial genome sequencing performed on DNA isolated from P3 muscle excluded pathogenic point mutations, with diagnostic long-range PCR assays and quantitative real-time PCR assays in single cells failing to detect mtDNA rearrangements. Clinical mitochondrial DNA analyses were not performed in P4-P8.

Levels of mtDNA from control and patient fibroblasts were subsequently analysed by Southern blot and qPCR, revealing a mild mtDNA depletion in P3, while in P1, P2, and P6 no significant depletion or deletions of mtDNA were observed (Fig. 3a). The effect of the _POLRMT_ variants on mitochondrial gene transcription was subsequently assessed using RT-qPCR (Fig. 3b). Transcript levels of six mitochondrial genes involved in OXPHOS activity were down-regulated in all four _POLRMT_ patients compared to control cells. Cells isolated from P1, P2 and P6 showed modest effects on gene expression (31–66%, 46–70% and 42–68% of wild-type levels, respectively), whereas cells isolated from P3 showed severe loss of mitochondrial transcripts (5–34% of wild-type levels). Thus, the identified _POLRMT_

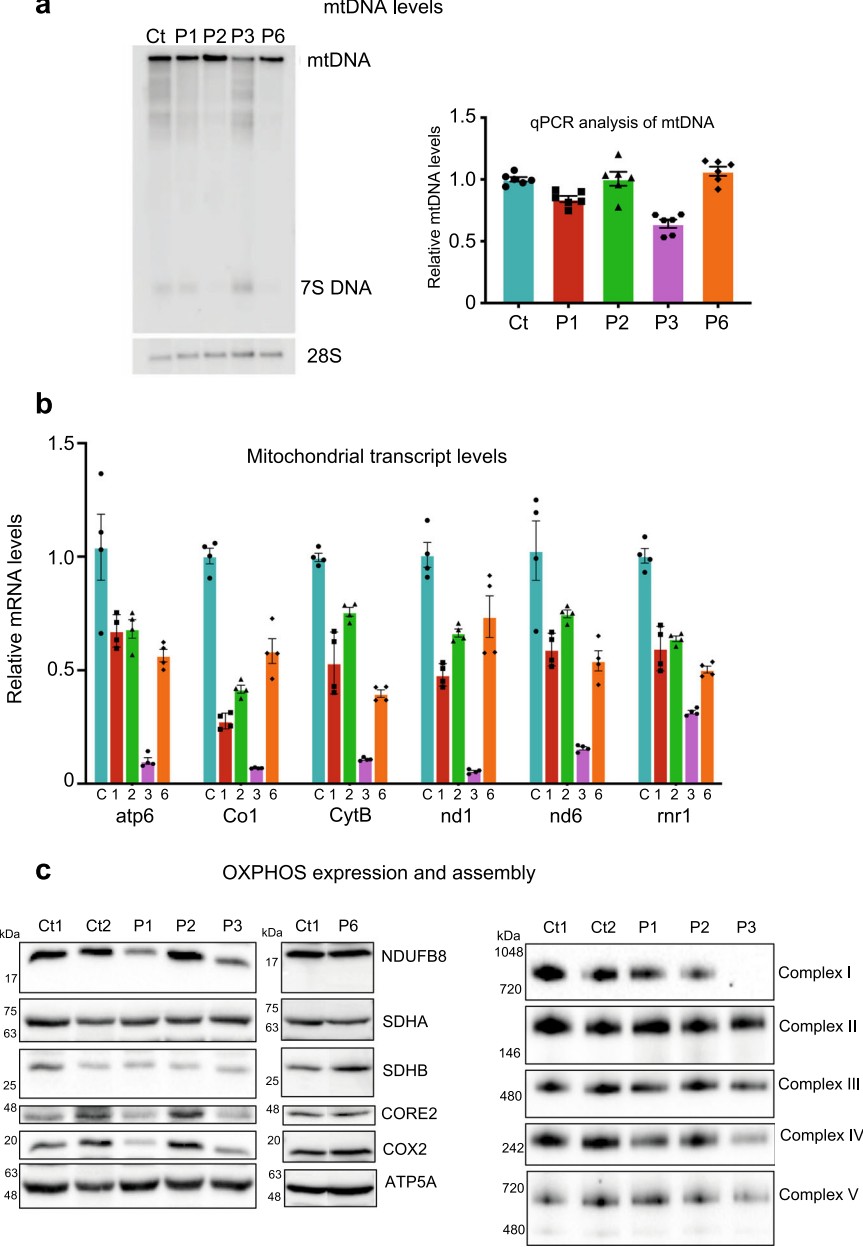

**Fig. 3 mtDNA, mitochondrial transcript, and OXPHOS protein levels in *POLRMT* patient fibroblasts. a** Effects of the patient variants on mtDNA levels in vivo were analysed by Southern blot (left panel) and qPCR (right panel). No significant deletions/depletion of mtDNA were observed in P1, P2, and P6 while mild mtDNA depletion was observed in P3 when compared to control (Ct). Southern blotting was repeated three times and a representative example is shown. For qPCR analysis, 6 technical repeats were performed (patient cells were grown in two separate cell cultures, which were each analysed three times). **b** Levels of mitochondrial transcripts in patient fibroblasts were analysed by RT-qPCR. P1, P2, and P6 showed a mild down-regulation of gene expression whereas P3 showed a severe loss of mitochondrial transcripts (5–32% of wild-type levels). For qPCR analysis, 4–6 technical repeats were performed (patient cells were grown in two separate cell cultures, which were each analysed 2–3 times). **c** Western blot analysis of whole-cell lysates extracted from adult control (Ct1, Ct2) and *POLRMT* patient (P1, P2, P3, P6) fibroblasts (left panel). P1 and P3 showed a decrease in the steady-state levels of subunits of CI (NDUFB8), CIII (UQCRC2), and CIV (COXII). Normal levels of OXPHOS subunits were detected in P2 and P6. One-dimensional BN PAGE analysis revealed a combined OXPHOS assembly defect in P3 and a slight defect in the assembly of CIV in P1 (right panel). The nuclear-encoded SDHA and/or SDHB subunits were used as loading controls. Representative blots of three independent SDS and BN PAGE experiments for P1, P2, and P3 are shown. One representative western blot experiment out of two independent analysis is shown for P6.

variants appear to exert their pathogenic effect by primarily influencing gene expression rather than mtDNA replication.

**Human *POLRMT* variants affect OXPHOS subunit synthesis and assembly in vivo.** To further characterise the effect of *POLRMT* variants on OXPHOS, western blot and blue native (BN) PAGE analyses were performed on protein extracts isolated

from control and patient primary dermal fibroblasts. In P1 and P3, we observed a decrease in the protein levels of subunits of Complex I (NDUFB8), Complex III (UQCRC2), and Complex IV (COXII), while normal levels of OXPHOS subunits were detected in P2 and P6 (Fig. 3c). One-dimensional BN PAGE analysis revealed a slight defect in the assembly of Complex IV in P1 (Fig. 3c). Consistent with normal steady-state levels of

mitochondrial OXPHOS subunits, no marked changes were observed in the nascent OXPHOS complexes in P2, except for a very mild decrease in Complex I (Fig. 3c). However, analysis of mitochondrial extracts isolated from P3 revealed a combined OXPHOS assembly defect affecting Complexes I, III, and IV (Fig. 3c). Interestingly, the loss of mitochondrial mRNA transcripts (Fig. 3c) and the OXPHOS assembly defect was more severe in P3, which may reflect the more deleterious nature of the dominant c.2641-1G > C, p.Gly881_Lys883del POLRMT variant.

**Disease-associated mutations impair mitochondrial transcription in vitro.** We next set out to characterise the enzymatic activities of the mutant forms of POLRMT in vitro. To this end, we expressed POLRMT in recombinant form, recapitulating the pathogenic mutations identified in all seven families (Fig. 4a). First, we investigated if the disease-causing mutations affect the thermal stability of POLRMT using differential scanning fluorimetry. We found that the thermal stability was relatively unaffected compared to wild-type (Supplementary Figure 2). We next monitored the ability of wild-type POLRMT and mutant derivatives to support promoter-dependent initiation of transcription. To this end, we used a linear template containing the mitochondrial light strand promoter and the reactions were performed in the presence of TFAM and TFB2M (Fig. 4b). Under these conditions, most of the individual mutations displayed severely reduced transcription levels. The exceptions were P556S/ S1193F, H250D and F641L, which only displayed mild reductions in activity (46–60% of wild-type activity).

Next, we repeated the transcription reactions, but now mixed the mutant POLRMT in the combinations present in patients. On a linear template, the mutant combinations found in patients P1, P2, P4, and P5 displayed an intermediate reduction in transcription levels (36–68% wild-type activity) while the P3, P6, and P7/8 combinations showed severe loss of transcription activity (3–13% wild-type activity) (Fig. 4c). A similar trend was observed when using a circular supercoiled template, with P1, P2, P4, and P5 showing milder transcription defects compared to P3, P6 and P7/8 (Supplementary Figure 3A).

To analyse POLRMT activity independently of the accessory transcription factors TFAM and TFB2M, we used a single nucleotide incorporation assay (Fig. 4d). In this assay, all the characterised disease combinations caused reduced nucleotide incorporation. The P1, P3, P4 and P5 combinations were associated with mild defects, while the P2, P6, and P7/8 variants showed severe loss of activity on a primed double-stranded template (Fig. 4d).

Finally, POLRMT is required for the synthesis of primers required for mtDNA replication at OriL. To monitor effects of this reaction, we used a template containing the OriL sequence in its active, single-stranded conformation (Fig. 4e) and analysed the activities of the different disease combinations found in patients (Fig. 4a). Primer synthesis was reduced in all mutant combinations analysed, with P2, P6, and P7/8 being particularly affected.

## Discussion

Defects in mtDNA maintenance genes are typically associated with PEO, encephalopathy, myopathy or hearing loss[16,20]. Similarly, to the mitochondrial-disease manifestations associated with POLG gene defects, human POLRMT variants cause a wide range of clinical symptoms which may manifest in childhood or adulthood. One adult patient (P3) from our cohort of eight patients harbouring POLRMT variants presented with an adult-onset PEO and two adult patients from the same family only manifested muscle weakness. However, the predominant clinical phenotype of the five paediatric patients harbouring POLRMT variants was an early onset moderate to severe developmental

delay, mild to severe intellectual disability, hypotonia/muscle weakness, short stature, and speech delay. The clinical presentation of POLRMT mutations is variable and there is no clear distinction between autosomal recessive and autosomal dominant disease. However, such clinical heterogeneity is well-described across a broad range of mitochondrial genetic aetiologies[1].

POLRMT is the sole mitochondrial primase and its deletion leads to a profound decrease in mtDNA replication that is associated with severe mitochondrial dysfunction and dilated cardiomyopathy in POLRMT knockout mice[21]. However, the mutant POLRMT variants studied here did not seem to dramatically affect either mtDNA copy number or length in patient skeletal muscle and fibroblasts (Fig. 3a). At low POLRMT levels in mice, transcription initiation from LSP is better maintained and an RNA primer for mtDNA replication at OriH is still synthesised[21]. This may also be the case for the patients analysed here, where POLRMT activity is compromised. In addition, even if in vitro analysis revealed reduced primer synthesis at OriL (Fig. 3e), the levels produced are apparently sufficient to maintain initiation of L-strand replication and mtDNA levels in vivo. Instead, the mutant POLRMT variants appear to primarily affect mRNA levels (Fig. 3b). These significant drops in transcript levels were only associated with mild respiratory chain defects in fibroblasts derived from P1 and P3 (Fig. 3c). It is possible that the half-life of mitochondrially-encoded mRNAs in POLRMT patient fibroblasts is prolonged in order to compensate for the decreased synthesis of mitochondrial transcripts[5]. It is important to note, however, that OXPHOS defects are often restricted to certain tissues and are not necessarily evident in cultured fibroblasts[21–23]. Indeed, POLRMT conditional knockout mice present with a severe combined defect in the mitochondrial respiratory chain enzymes in cardiac muscle[21]. Besides the two key components of mitochondrial nucleoids, TFAM and the mitochondrial single-stranded DNA-binding protein (mtSSB), a number of other mitochondrial proteins including POLRMT are prominently enriched in the nucleoprotein complex[24–26]. Therefore, we investigated the effect of POLRMT variants on the nucleoid distribution and dynamics of the mitochondrial network in patient fibroblasts. We did not observe any significant alterations in the mitochondrial network dynamics (Supplementary Figure 4) and the nucleoid size and distribution also appeared normal, however, no quantitative assessment was made (Supplementary Figure 4). These data suggest that the POLRMT variants studied here directly target mRNA synthesis without affecting nucleoid maintenance per se.

Our in vitro transcription data provide strong evidence that the RNA polymerase activity of the POLRMT variants is severely affected (Fig. 4). The patient variants studied here are distributed across three POLRMT domains (Fig. 2a). Each of these domains performs a particular function, suggesting that the effects of the mutations may be distinct and possibly synergistic and may be a reflection on the severity and spectrum of clinical phenotypes associated with these mutations. Analysis of the combinations of POLRMT mutations observed in patients revealed reduced transcription and in most cases, there was a similar reduction of activity in run-off transcription and single-nucleotide incorporation assays. A notable exception was P3. For this patient, we observed severe defects in run-off transcription activity, but the single-nucleotide incorporation levels were only mildly affected (P3). In this patient, a wild-type copy of POLRMT still exists but the Δ881–3 variant appears to exert a dominant negative effect over the wild-type. This is possibly a result of altered transcription dynamics due to lowered flexibility of the 881–883 region (due to loss of G881) and an inability to efficiently switch between initiation and elongation states, manifesting itself as impaired processive transcription activity (Supplementary Figure 3B and

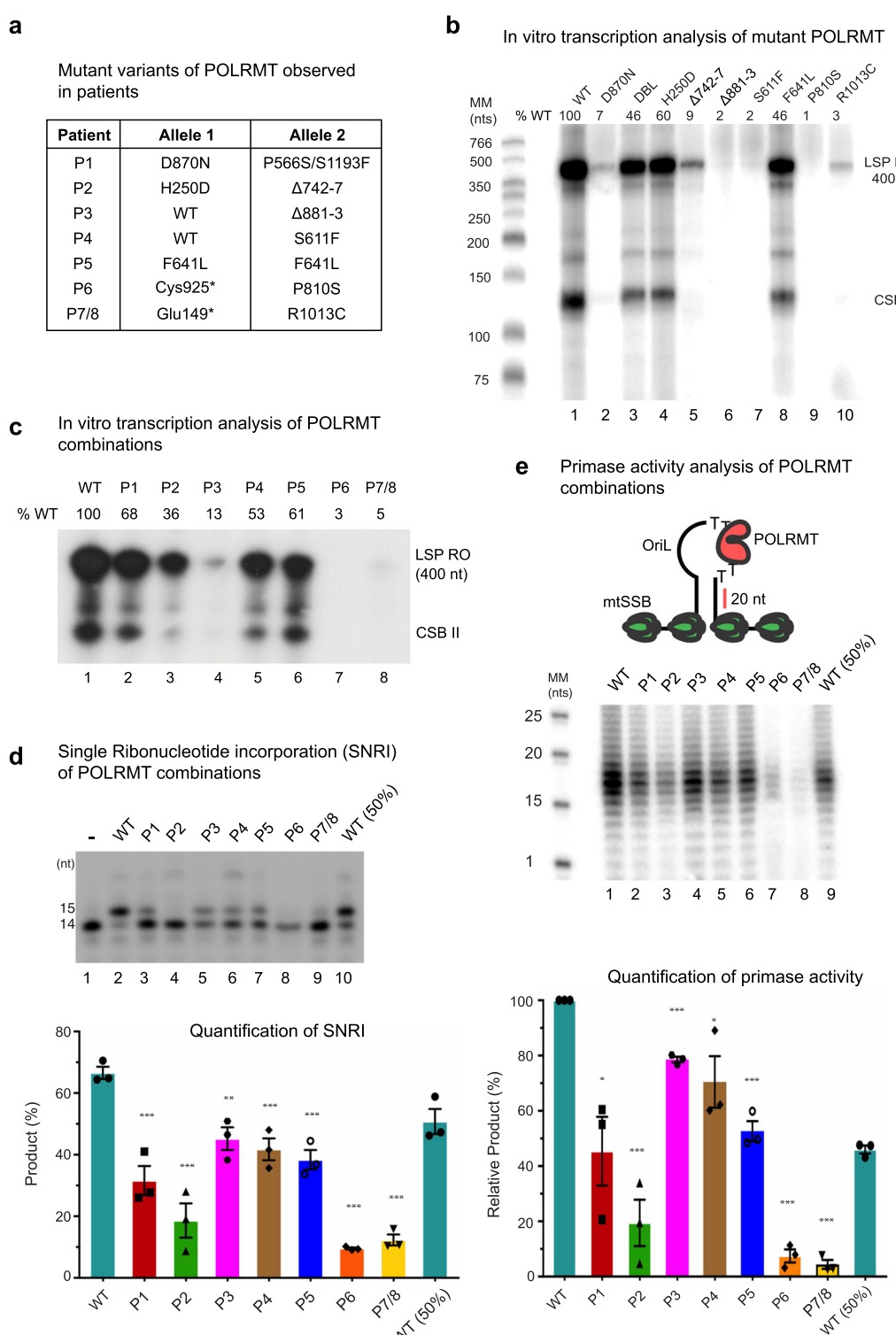

**Fig. 4 In vitro transcription assays of wild-type and mutant POLRMT. a** Combination of mutant alleles present in individual patients. **b** Promoter-dependent transcription from a linear LSP template comparing the activities of individual mutant POLRMT variants. The 400 nt run-off product (LSP RO) is indicated along with the pre-terminated transcription products formed at CSB II. Each POLRMT variant was analysed at least 3 times and a representative experiment is shown. **c** Promoter-dependent transcription as in panel **b**, but now using the combination of mutations identified in affected patients. The levels of run-off products relative to wild-type are indicated above each lane. Each POLRMT combination was analysed at least 3 times and a representative experiment is shown. **d** Single ribonucleotide incorporation (SNRI) assay comparing the combination of mutations identified in affected patients (top panel). Percentage of RNA product relative to the labelled substrate is indicated in the bar graph below. Data are presented as mean ± SEM of three individual experiments. Unpaired *t* test was performed to check significance. Asterisks represent statistically significant differences (*, *P* < 0.05; **, *P* < 0.01; ***, *P* < 0.005). **e** Primer synthesis on template containing the OriL stem loop structure with the indicated *POLRMT* variants (top panel) revealed mild to strong impairment of primase activity compared to the wild-type (middle panel). In the bar graph (lower panel), the percentage activity relative to wild-type is shown for each lane. Data are presented as mean ± SEM of three individual experiments. The *P* value was calculated as described in panel (**d**).

Supplementary Movie 1). Conformational flexibility, or a lack of, is therefore a possible contributor to the patient disease phenotype. For two of the mutants located in the C-terminal domain of the protein, P810S (P6) and R1013C (P7-8), we observed a near complete loss of transcription in vitro. We believe that the effect of these mutations is most likely exacerbated in vitro, possibly due to a lack of correct molecular chaperones or binding partners, since a complete loss of mitochondrial transcription would be incompatible with life.

Previous reports suggested that a *POLRMT* splice variant (spRNAP-IV) coordinates the transcription of nuclear genes[27]. However, it is now clear that POLRMT is solely responsible for the transcription of the mitochondrial genome, with evidence from expression and fluorescence imaging studies in human and mouse models[28]. Our data also support the exclusive role of POLRMT in mitochondrial transcription as demonstrated by the in vivo mitochondrial gene expression (Fig. 3b) and in vitro mitochondrial transcription studies (Fig. 4). The nuclear function for POLRMT is implausible, given the origin of the phage-related single-subunit mitochondrial RNA polymerase POLRMT, in contrast to the multi-subunit RNA polymerases (RNAPs) found in the eukaryotic nucleus, bacteria, and archaea[29]. Furthermore, structural studies revealing the molecular basis of mitochondrial transcription initiation demonstrated that the mechanism of POLRMT is distinct from the multi-subunit nuclear RNAPs, requiring specific transcription factors (TFAM and TFB2M) that structurally differ from their nuclear counterparts. Despite some topological similarities related to trapping of the non-template DNA strand found in nuclear and bacterial systems, TFAM and TFB2M have been shown to have specific roles in assisting POLRMT during promoter-dependent initiation[11]. Interestingly, mutations in *TFB2M*, were recently associated with an autistic spectrum disorder and developmental delay and shown to cause increased levels of transcription in patient cell lines[30]. It is therefore possible that mutations in other components of the mitochondrial transcription machinery could lead to effects related to those observed here with *POLRMT* mutations. Future investigations involving more patients are required to draw any clear conclusions.

In summary, we report the identification of pathogenic variants in the mitochondrial RNA polymerase, POLRMT, that underlie the mitochondrial disease-associated phenotypes present in the seven families we have studied. Our in vivo and in vitro functional studies establish *POLRMT* as a candidate disease gene that should be considered in the genetic diagnostic workup of patients that present with global developmental delay, speech/intellectual disabilities, hypotonia, short stature, PEO and multiple OXPHOS defects, and suggest that defects in the human mitochondrial transcription machinery may be more prevalent than previously thought.

## Methods

**Genetic, histopathological, and biochemical studies.** Massive parallel sequencing including whole exome, whole genome, or nuclear mitochondrial gene panel sequencing, followed by filtering and candidate variant analysis was undertaken in all eight patients studied. Details of the various sequencing strategies employed are provided in the supplementary appendix (Supplementary materials and methods 2.1). Diagnostic skeletal muscle biopsies from P1, P2, P3, P7, and P8 were obtained and subsequently processed for diagnostic histology and mtDNA molecular analyses according to standard procedures[19,31–35]. Large-scale mitochondrial DNA rearrangements in P3 muscle were screened by long-range PCR using a pair of primers (L6249: nucleotides 6249–6265; and H16125: nucleotides 16215–16196) to amplify a ~10 kb product in wild-type mitochondrial DNA (GenBank Accession number NC_012920.1)[33]. The diagnostic in vitro spectrophotometric assessment of respiratory chain complex activities (complexes I + III, complexes II + III, and complex IV) and citrate synthase were performed on P1, P2, and P3 muscle homogenates and P1, P2 fibroblasts[19,31,36].

**Cell culture conditions.** Primary *POLRMT* patient fibroblasts (P1, P2, P3, and P4) and controls (C1, C2) were cultured in high glucose Dulbecco's modified Eagle's medium (Sigma) supplemented with 10% fetal calf serum, 1 X non-essential amino acids (Sigma), 50 U/ml penicillin, 50 μg/ml streptomycin (Sigma) and 50 μg/ml uridine. The studied cell lines were mycoplasma free.

**DNA isolation and Southern blot analysis.** Total DNA was isolated from control and patient fibroblasts (2–4 ×10⁶ cells) using the High Pure PCR template preparation kit (Roche) according to the manufacturer's instructions. DNA (5 μg) was linearised by digestion with *Pvu*II and separated on a 0.8% agarose gel followed by overnight Southern blotting onto a nitrocellulose membrane. The DNA was UV-crosslinked to the membrane and hybridised with 32P-labelled probes corresponding to the D-loop or 28 S DNA sequences (see Supplementary Table 1) at 60 °C. Membranes were washed twice with 2X SSC buffer/0.1% SDS for 5 min at 60 °C, followed by two washes with 0.2X SSC buffer/0.1% SDS for 15 min at 60 °C. Signals were subsequently visualised and quantified using a FLA7000 Phosphoimager (Amersham) and MultiGauge software (Fujifilm).

**qPCR Quantification of mtDNA levels.** For qPCR analysis of the isolated total DNA, quantitative PCR reactions were performed using the indicated primer pairs to quantify both nuclear and mitochondrial DNA (see Supplementary Table 1). Amplification levels of mitochondrial DNA from all samples were normalised to their nuclear control and the results were plotted relative to the control fibroblast sample.

**RNA isolation and real-time qPCR analysis of mRNA transcript levels.** The levels of mitochondrial transcripts were assessed using real-time qPCR. Total RNA was isolated from control and patient fibroblasts (2–4 ×10⁶ cells) using the TRIzol reagent (Ambion) according to the manufacturer's instructions. RNA (2 μg) was DNAse-treated and used for cDNA synthesis using the High-Capacity cDNA Reverse Transcription kit (Invitrogen). Real-time qPCR was performed on the synthesised cDNA using a TaqMan Universal Master Mix II (ThermoFisher Scientific) and probes specific for mitochondrial transcripts (see Supplementary Table 1). Expression levels of the indicated mitochondrial transcripts were normalised to GAPDH and 18 S rRNA.

**Western blot and BN-PAGE analysis.** Protein analysis by Western blot and blue-native (BN) polyacrylamide gel electrophoresis (PAGE) was performed according to published methodologies[37,38]. Briefly, pelleted control and patient fibroblasts were resuspended in lysis buffer [50 mM Tris-HCl pH 7.5, 130 mM NaCl, 2 mM MgCl₂, 1 mM phenylmethanesulphonyl fluoride (PMSF), 1% Nonidet™ P-40 (v/v), and EDTA free protease inhibitor cocktail (Pierce)] and lysed for 30 min on ice, followed by centrifugation at 500 g for 5 min. The supernatant was retained, and Bradford method (Bio-Rad) was used to determine protein concentration. Equal amounts of protein samples resuspended in Laemmli sample buffer (1% SDS, 10% glycerol, 10 mM Tris-HCl, pH 6.8, 1 mM EDTA, and 50 mM dithiothreitol) were separated by sodium dodecyl sulphate–polyacrylamide gel electrophoresis (SDS-PAGE), followed by wet transfer to polyvinyl difluoride (PVDF) membrane (Immobilon™-P, Millipore Corporation) and subsequent probing with specific primary and HRP-conjugated secondary antibodies described above.

BN PAGE was performed on mitochondria isolated from cultured skin fibroblasts as in[37]. In brief, mitochondrial membranes were solubilised using 2 mg/mg protein n-dodecyl β-D-maltoside (DDM) on ice for 20 min. Protein extracts (150 μg) were separated on a 4–16% native polyacrylamide gradient gel according to the Novex® NativePAGE™ Bis-Tris Gel System instructions. Subsequently, the proteins were transferred to a PVDF membrane and subjected to standard immunological analysis of OXPHOS complexes as described below.

**Immunoblotting.** The following primary antibodies were used in this study: MS601 (ab110411, Abcam, 1:1000 dilution), NDUFB8 (ab110242, Abcam, 1:1000 dilution), SDHA (ab14715, Abcam, 1:2000 dilution), UQCRC2 (ab14745, Abcam, 1:1000 dilution), COXI (ab14705, Abcam, 1:1000 dilution), ATP5A (ab14748, Abcam, 1:1000 dilution), and β-actin (Cloud Clone Corp. CAB340Hu22, 1:10 000 dilution). HRP-conjugated mouse secondary antibody was used (DAKO, P0260, 1:2000 dilution), followed by detection using the ECL (GE Healthcare) and BioRad imaging system (Image Lab).

**In silico analysis and structural modelling.** The structures of the POLRMT initiation complex (PDB ID: 6ERQ), elongation complex (PDB ID: 4BOC), and POLRMT alone (PDB ID: 3SPA) were used to assess the structural implications of the patient variants. In silico mutagenesis was performed in PyMol V 1.3 (Schrodinger, LLC) and UCSF Chimera V 1.10.2.

**Expression and purification of recombinant POLRMT variants.** Recombinant POLRMT variants containing an N-terminal MBP tag and lacking the mitochondrial targeting sequence (MTS; residues 1–42) were cloned and expressed in ArcticExpress E. coli cells (Agilent Technologies). Site-directed mutagenesis producing human *POLRMT* variants with disease causing amino acid changes

(D870N, P566S/S1193F, H250D, Δ742–7, Δ881–3, S611F, F641F, P810S, and R1013C) was carried out using the QuikChange XL Lightning site-directed mutagenesis kit (Agilent Technologies) according to the manufacturer's instructions. The primers used are provided in Supplementary Table 1. Mutations were confirmed by sequencing. Please note, the nonsense variants (Cys925* and Glu149*) were not produced nor analysed.

Expression of the recombinant proteins in *E. coli* was induced by addition of 1 mM IPTG at 16 °C for 18 h. Cells were harvested at 5000 rpm, 10 min, 4 °C using a Beckman JLA 8.1000 rotor. Cells were resuspended in lysis buffer (0.5 M NaCl, 20 mM Hepes-OH pH 8.0, 10 mM ß-mercaptoethanol) containing 1× protease inhibitors (for all purifications the 100× stock of protease inhibitors contained 100 mM phenylmethylsulfonyl fluoride, 200 mM pepstatin A, 60 mM leupeptin and 200 mM benzamidine in 100% ethanol) and mixed with an Ultra-Turrax (Merck) to lower viscosity. The solution was centrifuged at 20,000 g, 45 min, 4 °C, and the supernatant was collected[6,39]. The extract was loaded onto a 5 mL HiTrap Heparin HP column (GE Healthcare) pre-equilibrated with Buffer A (25 mM Tris-HCl pH 8.0, 0.4 M NaCl, 10% (v/v) glycerol and 1 mM DTT) and eluted in a NaCl gradient (0.2–1.2 M NaCl in Buffer A). The MBP tag was removed by overnight incubation with 500 U TEV protease at 4 °C. Proteins were further purified on a HiLoad 16/60 Superdex 200 column (GE Healthcare) and 1 mL HiTrap SP HP (GE Healthcare) and dialysed against Buffer A.

Mitochondrial transcription factor A (TFAM), B2 (TFB2M), and mitochondrial transcription elongation factor (TEFM) were cloned as 6 x His-tagged fusion proteins into the vector pBacPAK9 (Clontech). Recombinant baculoviruses for the individual proteins were prepared as described in the BacPAK manual (Clontech) and used to infect *Spodoptera frugiperda* (*Sf*9) insect cells (Termo Fisher Scientific). Sf9 cells were grown in suspension and collected 60–72 h after infection. Infected cells were frozen in liquid nitrogen and thawed at 4 °C in a lysis buffer containing 25 mM Tris-HCl (pH 8.0), 20 mM 2-mercaptoethanol, and 1x protease inhibitors. Cells were incubated on ice for 20 min and then transferred to a Dounce homogeniser and lysed by 20 strokes of a tight-fitting pestle. NaCl was next added to a final concentration of 0.8 M and the homogenate was swirled gently for 40 min at 4 °C. Extracts were next cleared by centrifugation at 45,000 r.p.m. for 45 min at 4 °C using a Beckman TLA 100.3 rotor, followed by purification with HIS-Select Nickel Affinity Gel (Sigma Aldrich)[6,36]. After loading of protein extracts, the affinity gel was washed with 40 mM imidazole in buffer A (25 mM Tris–HCl pH 8.0, 0.4 M NaCl, 10% glycerol, 10 mM 2-mercaptoethanol, and 1× protease inhibitors). Proteins were eluted from the affinity gel with 250 mM imidazole in buffer A. The proteins were further purified over HiTrap Heparin and HiTrap SP column (GE-Healthcare). The HiTrap columns were equilibrated in a buffer with 0.2 M NaCl, 25 mM potassium phosphate pH 7.0, 10% glycerol, 0.5 mM EDTA, and 1 mM DTT (Purification Buffer-0.2). The proteins were eluted by a 15 ml gradient (0.2–1.2 M NaCl) using the buffer conditions. Purified proteins were dialysed against Purification Buffer-0.2 and frozen in liquid nitrogen.

**In vitro transcription assays**. Transcription activity was probed using either a supercoiled circular or *Bam*HI-linearised pUC18 plasmid containing the mitochondrial light strand promoter (LSP). Transcription reactions (25 μl) contained 10 mM Tris-HCl pH 8.0, 10 mM MgCl$_2$, 60 mM NaCl, 100 μg/mL BSA, 1 mM DTT, 4 U RNase inhibitor (New England Biolabs), 400 μM ATP, 150 μM GTP, 150 μM CTP, 10 μM UTP, 0.02 μM α$^{32}$P-UTP (3000 μCi/mmol), and 100 fmol circular or linear LSP template. Reactions were mixed and 500 fmol POLRMT variant, 500 fmol TFB2M, and 5 pmol TFAM were added on ice. In addition, 500 fmol TEFM was added to reactions containing circular supercoiled LSP template. Reactions were incubated at 32 °C for 30 min, stopped by the addition of stop buffer (10 mM Tris-HCl pH 8.0, 0.2 M NaCl, 1 mM EDTA and 100 μg/ml proteinase K) followed by incubation at 42 °C for 45 min. Transcription products were purified by ethanol precipitation, and the pellets were dissolved in 20 μl loading buffer (98% formamide, 10 mM EDTA, 0.025% (w/v) xylene cyanol FF and 0.025% (w/v) bromophenol blue). Samples were analysed on 8% (linear LSP) or 5% (circular LSP) denaturing polyacrylamide gels followed by exposure on photo film. All experiments were performed in triplicate and quantified using ImageJ software, version 1.53. For combination experiments, 250 fmol of each POLRMT variant was used, yielding 500 fmol total POLRMT per reaction.

A single nucleotide incorporation (SNRI) assay was used to assess POLRMT-mediated transcription initiation from a primed double-stranded template. A 14-mer RNA oligonucleotide was γ-$^{32}$P end-labelled using T4 polynucleotide kinase (PNK; Thermo Scientific) and annealed to a 28-mer non-template DNA oligonucleotide and its complementary template strand in 1:1 molar ratio (Supplementary Table 1). Reactions (20 μl) contained 20 mM Tris-HCl pH 8.0, 10 mM MgCl$_2$, 60 mM NaCl, 100 μg/mL bovine serum albumin (BSA), 1 mM DTT, 4 U RNase inhibitor (New England Biolabs), 100 μM ATP, 100 fmol end-labelled template and 1 pmol of indicated combination of POLRMT (WT and P1-P7). Reactions were incubated at 32 °C for 5 min and stopped by the addition of loading buffer (98% formamide, 10 mM EDTA, 0.025% (w/v) xylene cyanol FF and 0.025% (w/v) bromophenol blue). Samples were analysed on 12% denaturing polyacrylamide sequencing gels. All experiments were performed in triplicate and error bars ± SEM were calculated.

Primer synthesis assay was also used to assess POLRMT-mediated initiation from OriL. Primase reactions (25 μl) were set up as described above with the following changes: 1 pmol ssDNA oligonucleotide, 400 μM ATP, 150 μM CTP, 10 μM GTP, 150 μM UTP and 0.02 μM α-$^{32}$P GTP (3000 μCi/mmol) and 1 pmol of indicated combination of POLRMT (WT and P1-P7). Reactions were incubated at 32 °C for 30 min, stopped by the addition of loading buffer (98% formamide, 10 mM EDTA, 0.025% (w/v) xylene cyanol FF and 0.025% (w/v) bromophenol blue). Samples were analysed on 12% denaturing polyacrylamide sequencing gels. All experiments were performed in triplicate and error bars ± SEM were calculated.

**Statistics**. For qPCR quantification of mtDNA levels, reactions were performed in triplicate from two biological repeats and error bars ± SEM were calculated. For mtRNA quantification, qPCR reactions were performed in duplicate from two independent biological repeats and error bars ± SEM were calculated. For in vitro transcription activity assays, reactions were performed in triplicate, quantified using MultiGauge software (Fujifilm) and reported ± SEM.

**Study approval**. All procedures were in accordance with the ethical principles of the Declaration of Helsinki. Written patient consent was obtained, and all the studies were performed in agreement with the approved guidelines of local ethics committees of each institution that participated in this study with samples stored in the Newcastle Mitochondrial Research Biobank (NRES Committee North East - Newcastle & North Tyneside 1; 16/NE/0267). The online tool GeneMatcher facilitated collaboration between research centres (https://genematcher.org).

**Reporting summary**. Further information on research design is available in the Nature Research Reporting Summary linked to this article.

## Data availibility

The authors declare that the data supporting the findings of this study are available within the paper and its supplementary information files. Accession number for POLRMT cDNA used in the study is NM_005035.3. The structures of the POLRMT initiation complex (PDB ID: 6ERQ), elongation complex (PDB ID: 4BOC) and POLRMT alone (PDB ID: 3SPA) were sourced from PDB (https://www.rcsb.org). Diagnostic next generation sequencing data can be made available by the authors on request. Source data are provided with this paper.

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

## Acknowledgements

R.W.T. is supported by the Wellcome Centre for Mitochondrial Research (203105/Z/16/Z), Mitochondrial Disease Patient Cohort (UK) (G0800674), the Lily Foundation, the UK NIHR Biomedical Research Centre for Ageing and Age-related disease award to the Newcastle upon Tyne Foundation Hospitals NHS Trust, the MRC/EPSRC Molecular Pathology Node and the UK NHS Highly Specialised Service for Rare Mitochondrial Disorders of Adults and Children. E.W.S. was in receipt of a Medical Research Council (MRC) PhD studentship. C.M.G. and M.F. are supported by the Swedish Research Council (2013–3621 to M.F., 2012–2583 to C.M.G.), Swedish Cancer Foundation, the Knut and Alice Wallenberg foundation and grants from the Swedish state under the agreement between the Swedish government and the county councils, the ALF agreement to M.F. (ALFGBG-727491) and C.M.G. (ALFGBG-728151). S.K. was supported by the grant NV19-07-00136 from the Ministry of Health of the Czech Republic, acknowledge institutional program UNCE/MED/007 of the Charles University, the project LQ1604 NPU II from the Ministry of Education, Youth and Sports of the Czech Republic, and thanks to The National Center for Medical Genomics (LM2018132) for instrumental and methodologic support with the WES analyses. T.M. was supported by the grant NV19-07-00149 from the Ministry of Health of the Czech Republic and RVO:67985823 (to Institute of Physiology CAS); Z.K. was in receipt of Charles University PhD project GA UK 772119. K.Õ. and S.P. were supported by the Estonian Research Council grants PRG471 and PUTJD827. The Broad Center for Mendelian Genomics (UM1 HG008900) is funded by the National Human Genome Research Institute with supplemental funding provided by the National Heart, Lung, and Blood Institute under the Trans-Omics for Precision Medicine (TOPMed) program and the National Eye Institute. M.H.W. is supported by T32GM007748. P.B.K. is supported by NIH R01 NS080929 and the Children's Miracle Network. Exome sequencing and analysis were provided by the Broad Institute of Harvard and MIT Center for Mendelian Genomics (Broad CMG) and was funded by the National Human Genome Research Institute, the National Eye Institute, and the National Heart, Lung and Blood Institute grant NIH UM1 HG008900 to Daniel MacArthur and Heidi Rehm. We thank Daniel MacArthur and Anne O'Donnell-Luria at the Center for Mendelian Genetics of the Broad Institute for their contributions to exome sequencing and analysis. P.E.B. is supported by NIH NINDS RO1 NS08372. We thank Doug Turnbull and Andrew Schaefer for their long-term support and care of patient 3 described in this manuscript and Laura Bone for technical support. We would like to thank Dan Wright and family and Warren Lammert and Kathy Corkins for donations to the McMaster Neuromuscular Clinic which has supported the care of patient 2 described in this manuscript.

## Author contributions

M.O., B.P., Z.S., H.D.M., M.S., E.H., E.W.S., E.L.B. and J.J.C. carried out the experiments. S.K., T.M., V.S., H.H., A.J.B., K.L.M., S.B., Z.K. and A.P. performed the clinical and molecular investigations of family 1; M.T. and L.I.B. performed the clinical and molecular investigations of family 2; M.O., E.L.B., P.E.B., G.G., E.W.S. and R.W.T. performed the clinical and molecular investigations of family 3; K.N.W. and C.E.P. performed the clinical and molecular investigations of family 4; K.K., H.N. and H.R. performed the clinical and molecular investigations of family 5; K.O., M.H.W. and S.P. performed the clinical and molecular investigations of family 6; S.B.S., L.P., E.A.E., C.C.B., L.M.K. and P.B.K. performed the clinical and molecular investigations of family 7. M.O., B.P., Z.S., E.W.S., G.S.G., M.F., C.G. and R.W.T. analysed and interpreted the data. R.W.T., C.G. and M.F. designed and supervised the study and acquired funding. M.O., B.P., M.F., C.G. and R.W.T. wrote the manuscript. All authors contributed to the revision of the manuscript.

## Competing interests

The authors declare no competing interests.

## Additional information

[1]Wellcome Centre for Mitochondrial Research, Translational and Clinical Research Institute, The Medical School, Newcastle University, Newcastle upon Tyne NE2 4HH, UK. [2]Department of Medical Biochemistry and Cell Biology, University of Gothenburg, Gothenburg, Sweden. [3]Research Unit for Rare Diseases, Department of Pediatrics and Adolescent Medicine, First Faculty of Medicine, Charles University, Prague 120 00, Czech Republic. [4]Section on Nephrology, Wake Forest School of Medicine, Winston-Salem, USA. [5]Center for Cardiovascular and Pulmonary Research, Department of Pediatrics, Nationwide Children's Hospital, The Ohio State University College of Medicine, Columbus, USA. [6]Division of Endocrinology, Nationwide Children's Hospital, The Ohio State University College of Medicine, Columbus, USA. [7]Department of Bioenergetics, Institute of Physiology of the Czech Academy of Sciences, Prague, Czech Republic. [8]Department of Human Molecular Genetics, Max Planck Institute for Molecular Genetics, Berlin, Germany. [9]Institute of Human Genetics, University Medical Center of the Johannes Gutenberg University, Mainz, Germany. [10]Genetics Research Center, University of Social Welfare and Rehabilitation Sciences, Tehran, Iran. [11]Department of Pediatric and Medicines, Division of Neuromuscular and Neurometabolic Diseases, McMaster University Children's Hospital, Hamilton, Canada. [12]Division of Human Genetics, Cincinnati Children's Hospital Medical Center, Cincinnati, OH, USA. [13]Department of Pediatrics, University of Cincinnati College of Medicine, Cincinnati, OH, USA. [14]Department of Pediatrics, Cardiovascular Foundation of Colombia, Floridablanca, Colombia. [15]Department of Clinical Genetics, United Laboratories, Tartu University Hospital, Tartu, Estonia. [16]Department of Clinical Genetics, Institute of Clinical Medicine, University of Tartu, Tartu, Estonia. [17]Broad Institute of MIT and Harvard, Cambridge, MA, USA. [18]Divisions of Newborn Medicine and Genetics and Genomics, Department of Pediatrics, Boston Children's Hospital, Boston, MA, USA. [19]Department of Genetics, Yale University School of Medicine, New Haven, CT, USA. [20]Division of Pediatric Neurology, Department of Pediatrics, University of Florida College of Medicine, Gainesville, FL, USA. [21]Center for Mendelian Genomics, Broad Institute of MIT and Harvard, Cambridge, Massachusetts, USA. [22]Division of Genetics & Genomics, Boston Children's Hospital and Harvard Medical School, Boston, MA, USA. [23]Department of Molecular Genetics & Microbiology, and Department of Neurology, University of Florida College of Medicine, Gainesville, FL, USA. [24]Genetics Institute and Myology Institute, University of Florida, Gainesville, FL, USA. [25]Department of Molecular and Human Genetics, Baylor College of Medicine, Houston, TX, USA. [26]These authors contributed equally: Monika Oláhová, Bradley Peter. [27]These authors jointly supervised: Claes M. Gustafsson, Robert W. Taylor. ✉email: claes.gustafsson@medkem.gu.se; robert.taylor@ncl.ac.uk

