## [Peer Review File · Nature Communications]

Reviewer #1 (Remarks to the Author):

The manuscript by Oláhová and colleagues describe POLRMT variants that appear to cause mitochondrial diseases in eight patients. They identified recessive and dominant variants in the POLRMT gene. Where it could be tested, there was a defect in mitochondrial mRNA synthesis, but no mtDNA deletions or copy number abnormalities. In vitro characterization of the recombinant POLRMT mutants confirmed defects on mitochondrial transcription.

The manuscript is very straightforward and I do not have major issues with it. Its selling point is the importance of discovering POLRMT mutations causing human diseases in 7 different families. The phenotypes were interesting, and variable. They also found some surprises, such as no defect in mtDNA levels. The effect on OXPHOS proteins is also mild, and the authors argue that in tissues it may be more problematic. In fact, muscle of a patient showed a large number of COX-negative fibers.

I think the idea is that processivity in the mutant forms is decreased, so the primase activity is reduced but enough to prime replication. In this context, I wonder whether it would be interesting to analyze the in vitro assays as a ratio between the full length and the shorter "smear" transcript products.

Carlos Moraes

Reviewer #2 (Remarks to the Author):

Mitochondrial RNA polymerase is the key enzyme driving expression of the mitochondrial genome. While an important protein, its contribution to mitochondrial dysfunctions and neurological pathologies has never been demonstrated. The authors of this study screened the genomes of eight neurological disease patients for mutations in POLRMT and analyzed the pedigree of the patients and the inheritance pattern of the mutations. The biochemical data demonstrate a decreased level of the mitochondrial transcripts and a reduced expression of the oxidative phosphorylation complexes, consistent with the observed mitochondrial dysfunction. The manuscript presents an in silico analysis of the POLRMT mutations and their biochemical characterization using in vitro assays. The study advances our knowledge on the cause of mitochondrial dysfunctions and validates POLRMT as a potential therapeutic target. The analysis of the mutations in POLRMT and characterization of these variants are the strength of this study, however some of these experiments fall short of the standards accepted in the field of transcription. The authors are requested to address the following concerns and perform new experiments, as described below.

1. The term "intramolecular" refers to interactions within a molecule and therefore should not be combined with the words "protein-protein"
2. In silico analysis of the POLRMT mutants is an important part of this study as it validates the finding that this enzyme is responsible for the pathologies observed. Please move this analysis from the supplement into the manuscript body.
3. The following inaccuracies in the in silico interpretation of the mutations should be eliminated:
 - P1. Residue D870 makes a salt bridge with R822. Remove "makes several H-bonds with neighboring residues" as they are not side-chain specific.
 - P3. The region 881-883 in the "palm" subdomain remains the same in all published structures of POLRMT (RMSD < 1Å). This region is conserved in T7 RNAP, in which it also does not change upon transition from initiation to elongation. Therefore the statement that this region "is the site of extensive conformational changes during the transition between transcription and elongation" is incorrect. Remove the Supplemental Movie 1, as it is misleading. It would be sufficient to say that mutations in the region 881-883 perturb the integrity of the corresponding structural element.
 - P4. This mutation (S611) is close to TFB2M, not TFAM. Correct the label in the left panel (Fig 2, C) and the text of the manuscript. Please use the term "intercalating hairpin" instead of "β-hairpin" to conform to the literature.
 - P5. Label the α-helix as "thumb" in Fig 2, C (third panel from the left)
 - P7, P8. Each cycle of nucleotide addition and translocation of POLRMT along the DNA is

accompanied by movement of helix Y, which harbors the R1013 mutation. This mutation is nowhere close to TFAM and promoter DNA. Correct the text and remove the panel (second from the right in Fig, 2C) called "initiation".

4. It is difficult to figure out which particular POLRMT mutations were used in the experiment shown in Figure 4A,B. Apparently, these were combinations of the mutations observed in the patients. A small panel with a table in Figure 4 matching P1-P7 with the corresponding single or double mutants would be very helpful.

5. To characterize the mutants correctly, one should first test their catalytic activity using a primer extension (or single nucleotide incorporation) assay. The authors attempted this in the experiment shown in Supplemental Figure 3. As can be seen from the control lane, WT POLRMT was able to extend only ~10% of the primer. Considering that POLRMT was used in a 15 fold molar excess over the RNA-DNA scaffold, only ~0.6% of all POLRMT molecules were involved in the reaction. Thus, the data comparing the WT and the mutant enzymes are of very low confidence under these conditions.

The problem with this experiment is most likely due to the primer-template construct used. The resulting elongation complex does not mimic well the natural transcription elongation complex, as it lacks the downstream DNA duplex, the upstream DNA duplex, and the NT strand in the region of the RNA-DNA hybrid. The rationale for choosing such a construct is unclear, particularly since a well-characterized (both biochemically and structurally) RNA-DNA scaffold exists (doi: 10.1038/nsmb.2683). The authors should repeat these experiments using an appropriate RNA-DNA scaffold having all essential topological elements (upstream DNA, NT strand). The concentrations of RNA polymerase and nucleic acids should definitively exceed the elongation complex K_d (>0.2 μM). At equimolar concentrations of POLRMT and RNA-DNA scaffold nearly a 100% primer extension efficiency is expected for the WT enzyme. This experiment should be presented in the manuscript rather than in the Supplement.

6. The kinetic experiments presented in Figure S4 do not conform to the standards of the field and should be removed from the manuscript. An insufficiently described commercial system was used for measurements of K_m using an unnatural fluorescent analog of the substrate NTP. The description of the RNA-DNA scaffold used in these experiments was not provided in Methods or Supplemental Methods. Thus, the data provided in Table S3 are not credible. The low activity mutants are determined to have a dramatically higher affinity to the fluorescent substrate (or low K_m) as compared to WT POLRMT, which undermines confidence in these data. Since all the described mutations are located far away from the POLRMT NTP binding site, there is no need to include such kinetic experiments in this study.

7. Experiment shown in Figure 4B can be moved to the Supplement to conserve space.

8. The results of the experiment shown in Fig. 4C are open to interpretations.

First, an important control (WT POLRMT at 50% concentration) is missing.

Second, the wide range of the observed activity change is highly questionable. Considering that it must be difficult to measure the expression level of mutant POLRMT in vivo and to emulate the dominant or recessive phenotype of the mutations (authors' intention), I would suggest to eliminate this experiment from the manuscript altogether.

9. The experiment involving OriL transcription is insufficiently described. The sequence of the DNA oligonucleotide used is not provided. An irrelevant reference is provided to point to a previous study describing this experiment.

10. The manuscript states that "...a mild primase activity defect was observed in all variants compared to wild-type {polymerase}". This statement is inaccurate, as the P2 mutant shows a significant reduction in transcription efficiency. The experiment with OriL should be repeated with all mutants, for consistency, and quantification of the data should be presented.

11. The Supplementary Figure 1 presents data for P3, but the figure suggests P1.

12. Figure 1B - insufficient description, please provide label definition (squares, circles, etc). Same figure - in the family 2, the p.Pro742_Pro747del mutation is written as Gln740Argfs*106 under the pedigree.

13. In Supplementary Figure 3B, it is mentioned that a linear template was used. However, the length of the expected runoff (400 nt) does not match the length of the runoff indicated (3000 nt). Which template was used in the experiment?

14. Lane 372. The mutations are distributed among 3, not 4 domains of POLRMT

Reviewer #3 (Remarks to the Author):

I think this paper is rather straightforward and convincingly show that mutations in POLRMT can cause human mitochondrial disease. There are multiple families and alleles and the findings are backed up by strong and high quality in vitro biochemistry. Overall, very convincing and will be of quite some interest.

The main problem I have with the paper is that it could do a better job in setting the stage for the reader. As the paper is now, it is accessible for a specialist but will be hard for the general reader. In particular, the introduction should be improved to do a better job in setting the stage as quite a bit is known about the biochemical function of POLRMT.

Specific points:

1. Abstract. "although no mtDNA deletions" I think it is to be expected that decreased mtDNA transcription will not lead to mtDNA deletions, although mtDNA depletion could be expected if replication primer formation is reduced. Please reword.
2. Abstract. "decreased fidelity". There is not a single experiment in the paper addressing transcriptional fidelity. Should be reworded.
3. Intro, POLRMT undergoes a conformational switch from initiation and elongation and interacts with different factors in these phases to finally terminate. This transcription cycle is not well explained at all. See e.g. Hillen et al 2018 (PMID 30190598) for a nice overview. Please reword and cite this reference. I think it is important to introduce this properly early on as the authors later on classify mutations as affecting different stages in the transcription cycle.
4. Intro, line 124-125. "for the first time". Remove this expression as most scientific papers would be pointless if they did not describe findings for the first time.
5. Results line 268-274. In one of the mouse knockout papers referenced (22), it is quite clear that primer formation can be maintained at low POLRMT levels/activities. This finding is well in line with the findings in the POLRMT mutant patients that mtDNA replication (primer formation) is largely unaffected, whereas transcription for gene expression is reduced. Please explain this for the reader.
6. Discussion, line 361. POLRMT is classified as a core component of the nucleoid according to ref 25, but so are also VDAC, LRPPRC, SHMT2 and many other proteins listed in table 1 of this paper. Clearly this concept of the nucleoid is outdated and more recent work should be cited. Everyone seems to agree that TFAM and SSBP1 are the main DNA binding proteins organizing the nucleoid. This needs to be rewritten.
7. Discussion, line 364-365. The size and distribution of nucleoids is not done in a quantitative way (superresolution microscopy, image analysis etc) and is as it stands anecdotal. I think the authors should down-tune this to say e.g. that nucleoid size and distribution were apparently normal, but no quantitative assessment was made.
8. Discussion. The big elephant in the room that is not addressed at all in the discussion is the question whether some of the developmental phenotypes can be explained by a nuclear function for POLRMT. I think this unlikely for many reasons, but the topic should at least be discussed and the authors explain why this is an unlikely scenario. There is one important paper in Nature refuting previous claims that POLRMT has a nuclear role (PMID 25297440). This paper should be referenced and the authors should provide more arguments why a nuclear function is unlikely (phage-like machinery in mitos etc).

REVIEWER COMMENTS

Reviewer #1 (Remarks to the Author):

The manuscript by Oláhová and colleagues describe POLRMT variants that appear to cause mitochondrial diseases in eight patients. They identified recessive and dominant variants in the POLRMT gene. Where it could be tested, there was a defect in mitochondrial mRNA synthesis, but no mtDNA deletions or copy number abnormalities. In vitro characterization of the recombinant POLRMT mutants confirmed defects on mitochondrial transcription.

The manuscript is very straightforward and I do not have major issues with it. its selling point is the importance of discovering POLRMT mutations causing human diseases in 7 different families. The phenotypes were interesting, and variable. They also found some surprises, such as no defect in mtDNA levels. The effect on OXPHOS proteins is also mild, and the authors argue that in tissues it may be more problematic. In fact, muscle of a patient showed a large number of COX-negative fibers.

I think the idea is that processivity in the mutant forms is decreased, so the primase activity is reduced but enough to prime replication. In this context, I wonder whether it would be interesting to analyze the in vitro assays as a ratio between the full length and the shorter “smear” transcript products.

Author’s comment: We thank the reviewer for their positive response to the manuscript. The theory that defective processivity is a contributing factor to enzyme deficiency is very interesting and one which we have discussed for selected mutations - most notably the $\Delta 881-3$ deletion. Here we observed stalling at CSB II, which could not be rescued by increased TEFM, levels (see Supplementary Figure 3B). However, we did not see any clear correlation between the other mutations and transcription stalling. We are therefore not confident that a comparison of the full-length and shorter products will be beneficial. We have, however, included a short discussion of the possible role a processivity defect may have in the context of P1 (page 19):

Page 19: “This is possibly a result of altered transcription dynamics due to lowered flexibility of the 881-883 region (due to loss of G881) and an inability to efficiently switch between initiation and elongation states, manifesting itself as impaired processive transcription activity (Supplementary Figure 3B).”

Reviewer #2 (Remarks to the Author):

Mitochondrial RNA polymerase is the key enzyme driving expression of the mitochondrial genome. While an important protein, its contribution to mitochondrial dysfunctions and neurological pathologies has never been demonstrated. The authors of this study screened the genomes of eight neurological disease patients for mutations in POLRMT and analyzed the pedigree of the patients and the inheritance pattern of the mutations. The biochemical data demonstrate a decreased level of the mitochondrial transcripts and a reduced expression of the oxidative phosphorylation complexes, consistent with the observed mitochondrial dysfunction. The manuscript presents an in silico analysis of the POLRMT mutations and their biochemical characterization using in vitro assays. The study advances our knowledge on the cause of mitochondrial dysfunctions and validates POLRMT as a potential therapeutic target. The analysis of the mutations in POLRMT and characterization of these variants are the strength of this study, however some of these experiments fall short of the standards accepted in the field of transcription. The authors are requested to address the following concerns and perform new experiments, as described below.

1. The term "intramolecular" refers to interactions within a molecule and therefore should not be combined with the words "protein-protein"

Author's comment: We thank the reviewer for pointing this out and we have removed the term "protein-protein" to avoid confusion (page 12).

2. In silico analysis of the POLRMT mutants is an important part of this study as it validates the finding that this enzyme is responsible for the pathologies observed. Please move this analysis from the supplement into the manuscript body.

Author's comment: We agree with the reviewer and have moved this information back to the main text of the manuscript (pages 12-13).

3. The following inaccuracies in the in silico interpretation of the mutations should be eliminated:

- P1. Residue D870 makes a salt bridge with R822. Remove "makes several H-bonds with neighboring residues" as they are not side-chain specific.
- P3. The region 881-883 in the "palm" subdomain remains the same in all published structures of POLRMT (RMSD < 1Å). This region is conserved in T7 RNAP, in which it also does not change upon transition from initiation to elongation. Therefore the statement that this region "is the site of extensive conformational changes during the transition between transcription and elongation" is incorrect. Remove the Supplemental Movie 1, as it is misleading. It would be sufficient to say that mutations in the region 881-883 perturb the integrity of the corresponding structural element.
- P4. This mutation (S611) is close to TFB2M, not TFAM. Correct the label in the left panel (Fig 2, C) and the text of the manuscript. Please use the term "intercalating hairpin" instead of "β-hairpin" to conform to the literature.
- P5. Label the α-helix as "thumb" in Fig 2, C (third panel from the left)
- P7, P8. Each cycle of nucleotide addition and translocation of POLRMT along the DNA is accompanied by movement of helix Y, which harbors the R1013 mutation. This mutation is nowhere close to TFAM and promoter DNA. Correct the text and remove the panel (second from the right in Fig, 2C) called "initiation".

Author's comment: We are grateful for the reviewer's expert knowledge of the POLRMT structure and have incorporated all of the suggested changes into the manuscript. We have also removed Supplementary Movie 1 to avoid any confusion.

4. It is difficult to figure out which particular POLRMT mutations were used in the experiment shown in Figure 4A,B. Apparently, these were combinations of the mutations observed in the patients. A small panel with a table in Figure 4 matching P1-P7 with the corresponding single or double mutants would be very helpful.

Author's comment: We thank the reviewer for pointing this out and have now included a table showing the combination of mutation for each patient (see Figure 4A).

5. To characterize the mutants correctly, one should first test their catalytic activity using a primer extension (or single nucleotide incorporation) assay. The authors attempted this in the experiment shown in Supplemental Figure 3. As can be seen from the control lane, WT POLRMT was able to extend only ~10% of the primer. Considering that POLRMT was used in a 15 fold molar excess over the RNA-DNA scaffold, only ~0.6% of all POLRMT molecules were involved in the reaction. Thus, the data comparing the WT and the mutant enzymes are of very low confidence under these conditions. The problem with this experiment is most likely due to the primer-template construct used. The resulting elongation complex does not mimic well the natural transcription elongation complex, as it lacks the downstream DNA duplex, the upstream DNA duplex, and the NT strand in the region of the RNA-DNA hybrid. The rationale for choosing such a construct is unclear, particularly since a well-characterized (both biochemically and structurally) RNA-DNA scaffold exists (doi: 10.1038/nsmb.2683). The authors should repeat these experiments using an appropriate RNA-DNA scaffold having all essential topological elements (upstream DNA, NT strand). The concentrations of RNA polymerase and nucleic acids should definitively exceed the elongation complex K_d (>0.2 μM). At equimolar concentrations of POLRMT and RNA-DNA scaffold nearly a 100% primer extension efficiency is expected for the WT enzyme. This experiment should be presented in the manuscript rather than in the Supplement.

Author's comment: We have tested the catalytic activity of POLRMT and mutant derivatives using the RNA-DNA scaffold suggested by the reviewer. The efficiencies of the reactions are now considerably higher. The new data are presented in Figure 4D.

In the experiment presented (Figure 4D) we use a 10-fold excess of POLRMT (4 nM template, 40 nM POLRMT). For the reviewer's benefit, we also include a data demonstrating the effects of increasing POLRMT concentrations with constant concentrations of the RNA-DNA scaffold (Figure for reviewer 2 – please see appendix). Included is also an experiment with 2-fold excess of POLRMT (40 nM template, 80 nM POLRMT), which generates similar results as those presented in the manuscript. All experiments are repeated at least 3 times and we also include statistics.

6. The kinetic experiments presented in Figure S4 do not conform to the standards of the field and should be removed from the manuscript. An insufficiently described commercial system was used for measurements of K_m using an unnatural fluorescent analog of the substrate NTP. The description of the RNA-DNA scaffold used in these experiments was not provided in Methods or Supplemental Methods. Thus, the data provided in Table S3 are not credible. The low activity mutants are determined to have a dramatically higher affinity to the

fluorescent substrate (or low K_m) as compared to WT POLRMT, which undermines confidence in these data. Since all the described mutations are located far away from the POLRMT NTP binding site, there is no need to include such kinetic experiments in this study.

Author's comment: We agree with the reviewer that the kit used was insufficiently described. Subsequent requests to the manufacturer for more information have gone unanswered so we have removed this data from the manuscript.

7. Experiment shown in Figure 4B can be moved to the Supplement to conserve space.

Author's comment: We have followed the reviewer's suggestion. It is now supplementary figure 3A.

8. The results of the experiment shown in Fig. 4C are open to interpretations. First, an important control (WT PORLMT at 50% concentration) is missing. Second, the wide range of the observed activity change is highly questionable. Considering that it must be difficult to measure the expression level of mutant POLRMT in vivo and to emulate the dominant or recessive phenotype of the mutations (authors' intention), I would suggest to eliminate this experiment from the manuscript altogether.

Author's comment: We have followed the reviewer's suggestion. We have instead introduced a new figure 4B, in which we have analysed the transcription activities of the individual mutations on a linear template.

9. The experiment involving OriL transcription is insufficiently described. The sequence of the DNA oligonucleotide used is not provided. An irrelevant reference is provided to point to a previous study describing this experiment.

Author's comment: We apologise for the lack of information and have updated the methods section to include a more detailed explanation as well as added the OriL sequence to Supplementary Table 2.

10. The manuscript states that "...a mild primase activity defect was observed in all variants compared to wild-type {polymerase}". This statement is inaccurate, as the P2 mutant shows a significant reduction in transcription efficiency. The experiment with OriL should be repeated with all mutants, for consistency, and quantification of the data should be presented.

Author's comment:

We agree with the reviewer and have repeated this experiment three times and also provide statistics. The data clearly indicate that the P2 mutant has reduced primase activity. Please see figure 4E in the new version of the manuscript.

11. The Supplementary Figure 1 presents data for P3, but the figure suggests P1.

Author's comment: We thank the reviewer for pointing this out. The Supplementary Figure 1 has been now amended and shown that these data relate to P3.

12. Figure 1B - insufficient description, please provide label definition (squares, circles, etc).

Same figure - in the family 2, the p.Pro742_Pro747del mutation is written as Gln740Argfs*106 under the pedigree.

Author's comment: Apologies for this mistake, the pPro742_Pro747del variant is now indicated in Family 2. We also updated the Figure legend 1B and provided detailed label definition.

“(B) Family pedigrees and segregation status of POLRMT patients. Affected individuals are depicted in black, circles represent females, squares represent males; a diagonal line through the symbol represents deceased individuals and double horizontal lines indicate consanguinity.”

13. In Supplementary Figure 3B, it is mentioned that a linear template was used. However, the length of the expected runoff (400 nt) does not match the length of the runoff indicated (3000 nt). Which template was used in the experiment?

Author's comment: Indeed, the length of the expected run off product was 400 nt and as indicated in the figure legend a linear LSP template was used. We have now amended the figure and stated the correct size of the run off product (400 nt).

14. Lane 372. The mutations are distributed among 3, not 4 domains of POLRMT

Author's comment: Apologies for this oversight, this has been now amended accordingly (page 18).

Reviewer #3 (Remarks to the Author):

I think this paper is rather straightforward and convincingly show that mutations in POLRMT can cause human mitochondrial disease. There are multiple families and alleles and the findings are backed up by strong and high quality in vitro biochemistry. Overall, very convincing and will be of quite some interest.

The main problem I have with the paper is that it could do a better job in setting the stage for the reader. As the paper is now, it is accessible for a specialist but will be hard for the general reader. In particular, the introduction should be improved to do a better job in setting the stage as quite a bit is known about the biochemical function of POLRMT.

Author's comment: We would like to thank the reviewer for his/her time and insightful comments. We have now expanded the introduction and included the description of the different stages of mitochondrial transcription and the role of POLRMT in this process. We have addressed each point raised by the reviewer below.

Specific points:

1. Abstract. “although no mtDNA deletions” I think it is to be expected that decreased mtDNA transcription will not lead to mtDNA deletions, although mtDNA depletion could be expected if replication primer formation is reduced. Please reword.

Author's comment: We acknowledge the reviewer's comment and reworded the abstract accordingly.

“Where investigated, patient fibroblasts revealed a defect in mitochondrial mRNA synthesis, and no mtDNA deletions or copy number abnormalities were identified.”

2. Abstract. “decreased fidelity”. There is not a single experiment in the paper addressing transcriptional fidelity. Should be reworded.

Author’s comment: We thank the reviewer for pointing this out and reworded the sentence to state that defective mitochondrial transcription drives the disease mechanism.

“Our *in vivo* and *in vitro* functional studies of POLRMT variants, establish defective mitochondrial transcription as an important disease mechanism.”

3. Intro, POLRMT undergoes a conformational switch from initiation and elongation and interacts with different factors in these phases to finally terminate. This transcription cycle is not well explained at all. See e.g. Hillen et al 2018 (PMID 30190598) for a nice overview. Please reword and cite this reference. I think it is important to introduce this properly early on as the authors later on classify mutations as affecting different stages in the transcription cycle.

Author’s comment: We have now expanded the introduction, covering the different stages of transcription cycle and highlighting the interactions between POLRMT and the different factors of the transcription machinery (pages 5-6). We included new reference and the review (PMID: 30190598 – reference 12) suggested by the reviewer.

“Large-scale conformational changes occur in these domains as the protein binds TFAM and mtDNA during transcription initiation as well as during the handover to processive mtDNA transcription⁹⁻¹². The initiation of transcription begins with TFAM binding the promoter upstream of the transcription start site and inducing a stable U-bend in the mtDNA. Following the recruitment of POLRMT a closed pre-initiation complex is formed. The interaction between POLRMT with TFAM is mediated via the N-terminal extension domain of POLRMT, also known as the ‘tether helix’, which is important for anchoring the active site of POLRMT near the transcription start site where the initial DNA melting occurs¹¹. Following the binding of TFB2M to the duplex DNA, TFB2M induces structural changes in POLRMT that stabilize the open DNA region. TFB2M contacts the conserved intercalating hairpin within the N-terminal domain of POLRMT, that results in melting of the DNA duplex and formation of the open initiation complex, where *de novo* RNA synthesis can begin¹¹. The transition from transcription initiation to elongation involves the release of TFB2M and recruitment of the mitochondrial transcription elongation factor (TEFM). TEFM forms a homodimer via its C-terminus and stabilizes interactions between the intercalating hairpin of POLRMT and the DNA template, thus separating the nascent RNA¹³⁻¹⁵. This interaction also contributes to the formation of a tight RNA exit tunnel along POLRMT, that is thought to enhance the processivity of the RNA polymerase along the DNA strand, and bypass premature termination or stalling caused by secondary structured RNA regions such as the G-quadruplex sequence of the conserved sequence block II (CSBII)¹³⁻¹⁵. In the absence of TEFM, the LSP-driven transcription is often terminated at the CSBII producing a 120-nt transcript, which after processing can be used by the replisome machinery to initiate replication at OriH⁴.”

4. Intro, line 124-125. “for the first time”. Remove this expression as most scientific papers would be pointless if they did not describe findings for the first time.

Author’s comment: We have now amended the sentence as recommended by the reviewer (page 7).

“Our data demonstrate that defective POLRMT can lead to classical mitochondrial disease and wider neurological manifestations which are independent of disordered mtDNA maintenance.”

5. Results line 268-274. In one of the mouse knockout papers referenced (22), it is quite clear that primer formation can be maintained at low POLRMT levels/activities. This finding is well in line with the findings in the POLRMT mutant patients that mtDNA replication (primer formation) is largely unaffected, whereas transcription for gene expression is reduced. Please explain this for the reader.

Author’s comment: We thank the reviewer for pointing this out. It is indeed the case that mtDNA replication appears to be largely unaffected despite the severe transcription defects observed. In Kuhl *et al.* (2016), it was shown that at low levels of POLRMT, transcription initiation from LSP was better maintained compared to HSP. Since transcription from both LSP and HSP is required for mtDNA gene expression, low levels of POLRMT (or lower POLRMT activity in the case of the mutant proteins) would favour preservation of mtDNA replication over transcription. We have included a short explanation of this in the discussion (page 17):

“At low POLRMT levels in mice, transcription initiation from LSP is better maintained and an RNA primer for mtDNA replication at OriH is still synthesized²³. This may also be the case for the patients analysed here, where POLRMT activity is compromised. In addition, even if *in vitro* analysis revealed reduced primer synthesis at OriL (**Figure 3E**), the levels produced are apparently sufficient to maintain initiation of L-strand replication and mtDNA levels *in vivo*.”

6. Discussion, line 361. POLRMT is classified as a core component of the nucleoid according to ref 25, but so are also VDAC, LRPPRC, SHMT2 and many other proteins listed in table 1 of this paper. Clearly this concept of the nucleoid is outdated and more recent work should be cited. Everyone seems to agree that TFAM and SSBP1 are the main DNA binding proteins organizing the nucleoid. This needs to be rewritten.

Author’s comment: We acknowledge the reviewer’s comment and have rewritten this section accordingly (page 18). We also included additional references.

²⁶ Farge and Falkenberg 2019 IJMS: Organization of DNA in Mammalian Mitochondria.

²⁸ Rajala et al. 2015 PLoS One: Whole cell formaldehyde cross-linking simplifies purification of mitochondrial nucleoids and associated proteins involved in mitochondrial gene expression.

“Besides the two key components of mitochondrial nucleoids, TFAM and the mitochondrial single-stranded DNA-binding protein (mtSSB), a number of other mitochondrial proteins including POLRMT are prominently enriched in the nucleoprotein complex²⁶⁻²⁸.”

7. Discussion, line 364-365. The size and distribution of nucleoids is not done in a

quantitative way (superresolution microscopy, image analysis etc) and is as it stands anecdotal. I think the authors should down-tune this to say e.g. that nucleoid size and distribution were apparently normal, but no quantitative assessment was made.

Author's comment: We thank the reviewer for their comment. We have rewritten this section as requested (page 18).

“Therefore, we investigated the effect of *POLRMT* variants on the nucleoid distribution and dynamics of the mitochondrial network in patient fibroblasts. We did not observe any significant alterations in the mitochondrial network dynamics (Supplementary Figure 4) and the nucleoid size and distribution also appeared normal, however, no quantitative assessment was made (Supplementary Figure 4). These data suggest that the *POLRMT* variants studied here directly target mRNA synthesis without affecting nucleoid maintenance *per se*.”

8. Discussion. The big elephant in the room that is not addressed at all in the discussion is the question whether some of the developmental phenotypes can be explained by a nuclear function for POLRMT. I think this unlikely for many reasons, but the topic should at least be discussed and the authors explain why this is an unlikely scenario. There is one important paper in Nature refuting previous claims that POLRMT has a nuclear role (PMID 25297440). This paper should be referenced and the authors should provide more arguments why a nuclear function is implausible (phage-like machinery in mitos etc).

Author's comment: We agree with the reviewer that this is an important study and expanded the discussion providing evidence on why a nuclear function for POLRMT is implausible. Please see modified text (page 19):

“Previous reports suggested that a POLRMT splice variant (spRNAP-IV) coordinates the transcription of nuclear genes²⁹. However, it is now clear that POLRMT is solely responsible for the transcription of the mitochondrial genome, with evidence from expression and fluorescence imaging studies in human and mouse models³⁰. Our data also support the exclusive role of POLRMT in mitochondrial transcription as demonstrated by the *in vivo* mitochondrial gene expression (Figure 3B) and *in vitro* mitochondrial transcription studies (Figure 4). The nuclear function for POLRMT is implausible, given the origin of the phage-related single-subunit mitochondrial RNA polymerase POLRMT, in contrast to the multi-subunit RNA polymerases (RNAPs) found in the eukaryotic nucleus, bacteria and archaea³¹. Furthermore, structural studies revealing the molecular basis of mitochondrial transcription initiation demonstrated that the mechanism of POLRMT is distinct from the multi-subunit nuclear RNAPs, requiring specific transcription factors (TFAM and TFB2M) that structurally differ from their nuclear counterparts. Despite some topological similarities related to trapping of the non-template DNA strand found in nuclear and bacterial systems, TFAM and TFB2M have been shown to have specific roles in assisting POLRMT during promoter-dependent initiation¹¹.”

Appendix

Figure for reviewer 2

Template 4nM: POLRMT 40nM
Figure 4D in the manuscript

Template 4nM: POLRMT 40nM
Figure 4D in the manuscript

Reviewer #2 (Remarks to the Author):

This revision satisfactory addressed all my suggestions and comments.

Reviewer #3 (Remarks to the Author):

The authors have nicely responded to my comments and I am now happy with this manuscript.